# Chemical instability at chalcogenide surfaces impacts chalcopyrite devices well beyond the surface

Diego Colombara [1,2,3,9✉], Hossam Elanzeery [1,4,9✉], Nicoleta Nicoara [2], Deepanjan Sharma[2], Marcel Claro [2], Torsten Schwarz[5], Anna Koprek[5], Max Hilaire Wolter[1], Michele Melchiorre [1], Mohit Sood [1], Nathalie Valle[6], Oleksandr Bondarchuk [2], Finn Babbe [1,7], Conrad Spindler[1], Oana Cojocaru-Miredin[5,8], Dierk Raabe[5], Phillip J. Dale [1], Sascha Sadewasser [2] & Susanne Siebentritt[1]

The electrical and optoelectronic properties of materials are determined by the chemical potentials of their constituents. The relative density of point defects is thus controlled, allowing to craft microstructure, trap densities and doping levels. Here, we show that the chemical potentials of chalcogenide materials near the edge of their existence region are not only determined during growth but also at room temperature by post-processing. In particular, we study the generation of anion vacancies, which are critical defects in chalcogenide semiconductors and topological insulators. The example of $CuInSe_2$ photovoltaic semiconductor reveals that single phase material crosses the phase boundary and forms surface secondary phases upon oxidation, thereby creating anion vacancies. The arising metastable point defect population explains a common root cause of performance losses. This study shows how selective defect annihilation is attained with tailored chemical treatments that mitigate anion vacancy formation and improve the performance of $CuInSe_2$ solar cells.

[1] Physics and Materials Science Research Unit, University of Luxembourg, Belvaux L-4422, Luxembourg. [2] International Iberian Nanotechnology Laboratory, Av. Mestre Jose Veiga, Braga 4715-330, Portugal. [3] Università degli Studi di Genova, via Dodecaneso 31, Genova 16146, Italy. [4] Avancis, Otto-Hahn-Ring 6, 81739 München, Germany. [5] Max-Planck-Institut für Eisenforschung GmbH, Max-Planck-Strasse 1, 40237 Düsseldorf, Germany. [6] Luxembourg Institute of Science and Technology, Belvaux L-4422, Luxembourg. [7] Joint Center for Artificial Photosynthesis, Lawrence Berkeley National Laboratory, 1 Cyclotron Road, 94720 Berkeley, CA, USA. [8] Institute of Physics, RWTH Aachen University, Sommerfeldstrasse 14, 52062 Aachen, Germany. [9] These authors contributed equally: Diego Colombara, Hossam Elanzeery. ✉email: diego.colombara@bath.edu; hossam.elanzeery@gmail.com

Atoms in a crystalline structure align in a regular lattice, but due to off-stoichiometry, thermal energy, reactions or phase changes, some of the atoms leave their lattice sites or fail to occupy them, generating point defects. The density of these defects (such as vacancies, antisites and interstitials) and their charge state (positive, negative and neutral) depend on the (electro)chemical potentials of the constituent atoms and electrons. These potentials are usually controlled by the elemental compositions during the growth of a material and are vital in shaping its properties.

Indeed, significant property changes can be observed with slight modifications of the synthesis conditions, especially when crossing the boundaries of phase homogeneity regions, as epitomized by the assorted realm of steels[1]. Therefore, a deliberate positioning along the edges of single-phase existence regions during growth can be exploited to benefit from the characteristics of the different phases involved, such as a superior native doping or microstructure.

However, these advantages may come at a cost. The growth-dependent (electro)chemical potential of the constituent atoms is a source of interface instability. Different interface reactivity manifests itself as undesirable chemical reactions affecting the material obtained on either side of the phase boundary. Furthermore, the detrimental nature of the reactions may spring from the formed secondary phase or from the altered defect population at the interface or both.

Understanding the nature of the defects involved, their concentration and mobility during the growth and after subsequent interface reactions is essential for the advancement of many technologies. For example, it is valuable for the control of the physical properties and functionality of chalcogenides in today's optoelectronic and future spintronic devices based on two-dimensional (2D) and three-dimensional (3D) semiconductors[2–4] and topological insulators[5]. Likewise, progress in the performance stability of perovskite-based photovoltaics (PVs) relies on strategies aimed at minimizing the formation, mobility or reactivity of anion vacancies[6].

Here, the case is made for $CuInSe_2$ (CIS), a 3D chalcogenide belonging to the adamantine family of materials with a wide single-phase existence region (Fig. 1a)[7,8]. CIS is a suitable proxy for the successful $Cu(In,Ga)(S,Se)_2$ (CIGS) PV technology and will become increasingly important for the third-generation concepts, e.g., future tandem cells[9] in combination with wider-gap chalcopyrites or halide perovskites[10,11].

Realizing today's highly efficient CIGS, solar cells require a carefully conceived fabrication process, one that derives from a decades-long research endeavour[12]. One key innovation in CIGS fabrication was enabled in the 1990s by crossing the phase homogeneity boundary during growth from 'Cu-poor' to 'Cu-rich', then back again to Cu-poor compositions (known as the three-stage process)[13].

The strategy received widespread success, because it allows combining the superior microstructure of Cu-rich material (cf. cross-sections in Fig. 1b) and the superior performance typical for Cu-poor compositions.

However, besides showing larger grains, Cu-rich CIS displays a more ideal luminescence signature than Cu-poor CIS, a prerequisite for high efficiency potential (Fig. 1d)[14]. This fact has puzzled the community for a long time, because it is at odds with the worse performance of Cu-rich CIS devices.

It is well known that during CIGS solar cell manufacturing, exposure of the absorber surface to ambient air[15], alkali metal fluorides[16,17] or chemical etchants[18,19] before buffer deposition influences heavily the device's optoelectronic properties. It is likely, but has never been investigated, that CIGS grown under different conditions displays different resilience to point defect formation during and after growth. It is unknown how the altered defect populations influence the PV performance in commonly processed CIGS absorbers from this fundamental viewpoint.

Here we identify the root causes of chemical instability of CIS when crossing the edge of its existence region and relate them to device's performance losses. Three thin-film compositions with increasing Cu concentration are investigated as bare absorbers, after surface treatments and as PV cells. These are identified

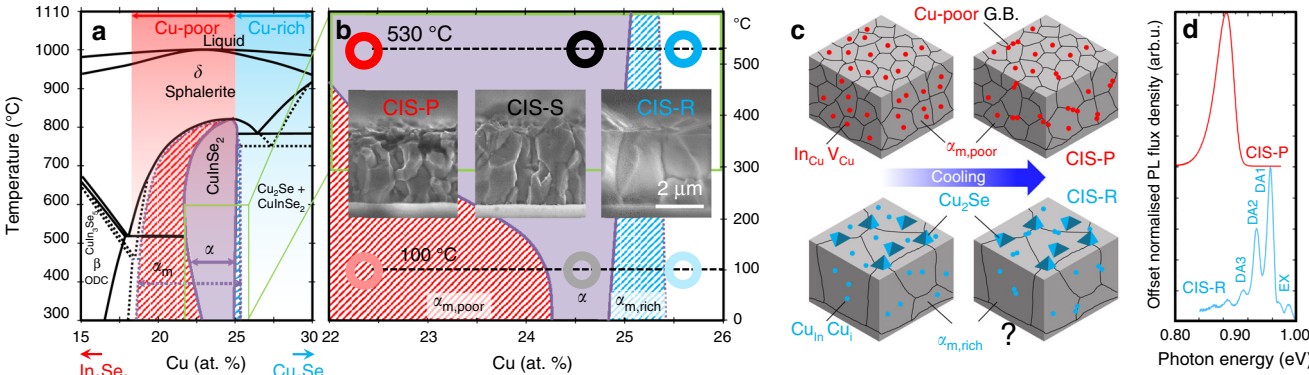

**Fig. 1 CuInSe₂ phase equilibria and microstructure.** **a** Portion of the $In_2Se_3$-$Cu_2Se$ pseudobinary diagram showing the stable (solid lines) and metastable (dotted lines) phase equilibria (reproduced with permission from Gödecke et al.[8] © Carl Hanser Verlag GmbH & Co.KG, München). The thin-film growth regions commonly referred to as Cu-poor and Cu-rich appear at the top as red and cyan fading portions, respectively. **b** Magnification of the phase diagram in the region of interest for CIS processing (and extension to room temperature[8]) with hollow circles indicating the compositions investigated: CIS-P (red, Cu-poor), CIS-S (black, nearly stoichiometric) and CIS-R (cyan, Cu-rich). Insets: corresponding cross-sectional scanning electron microscopy images (CIS-R subject to KCN etching to remove the excess $Cu_2Se$ phase). **c** Schematic illustration of Cu-poor (top) and Cu-rich (bottom) films after growth (left) and cool down (right). In the Cu-poor case, $In_{Cu}$ antisites are dispersed within the CIS lattice and part of them accumulate at grain boundaries during cool down, leading to Cu-poor grain boundaries (GB). Typical compositions of grain interiors at room-temperature match with the metastable Cu-poor region of existence, red patterned region in **b**. Similarly, $Cu_{In}$ antisites are dispersed within the CIS lattice of Cu-rich material, but Cu interstitials may also be present, possibly accounting for the enhanced grain growth during processing and for the existence of a supersaturated metastable phase at room temperature, cyan patterned region in **b**. **d** Photoluminescence spectra of CIS-P and CIS-R films acquired at 10 K. Whereas CIS-R shows excitonic emissions (labelled EX, indicative of high crystal quality) and up to three donor–acceptor pair transitions (labelled DA), CIS-P shows no excitonic emissions but broader, more red-shifted and more asymmetric defect-related transitions, typical of a high degree of compensation[14].

**Table 1 Sample identification legend.**

**CIS growth compositions**

| Details | Naming |
|---|---|
| Cu-poor, red colour | P (as Poor) |
| Nearly stoichiometric, black colour | S (as Stoichiometric) |
| Cu-rich, blue colour | R (as Rich) |

**CIS surface post deposition treatments (PDTs)**

| Details | Naming |
|---|---|
| Unetched, as grown | U (as Unetched) |
| 30 s KCN 5 % wt. at room temperature | W (as Weak etch) |
| 300 s KCN 10% wt. at room temperature | B (as-Bold etch) |
| ≤480 s under Se flux ≤ 250 °C, ref. [60] | Se-only |
| ≤480 s under Se and KF fluxes ≤ 390 °C, ref. [70] | Se/KF |
| ≤480 s under Se and In fluxes ≤ 300 °C ref. [57,58] | Se-In |
| Thiourea-rich chemical bath deposition (CBD) of Zn(O,S) onto the surface of CIS, ref. [60] | CBD-Zn(O,S) |

Combinations of CIS types and surface treatment type are indicated by combining P, S or R with U, W or B, accordingly.

according to the naming in Table 1 with associated colour coding in Fig. 1b and throughout the figures. The films are grown via a one-stage co-evaporation process at 530 °C, followed by unforced cooling to room temperature (as for conventional lab-scale thin-film processing). Our results show that amphoteric selenium-copper divacancies can form at the surface of both Cu-poor and Cu-rich CIS at room temperature, albeit at different extent. In Cu-poor material, whereas wet chemical bath processing may be enough to mitigate the adverse effects of the divacancy, vacuum processing may not. In Cu-rich material, passivation of the divacancy is more difficult but can be achieved under chalcogen excess.

## Results

**Point defect solubility limits and phase metastability.** Throughout the manuscript, the word 'metastability' refers to a sample condition describing a stable state of a dynamical system other than the system's state of least energy. Depending on the context, the term applies both to phase equilibria and electronic defect states.

The starting point for discussion is the $In_2Se_3$-$Cu_2Se$ pseudobinary system, as determined by Gödecke et al.[8]. Figure 1a illustrates the portion of phase diagram in the range 15–30 atomic % (at.%) Cu, showing the CIS homogeneity region of existence ($\alpha$). The edges of the homogeneity region under equilibrium conditions are indicated by the purple solid lines and area. These edges were determined from the analysis of bulk material subject to step cooling, i.e., alternating slow cooling periods (2 °C min$^{-1}$) and periods at constant temperature. On the contrary, water-quenching from the sphalerite $\delta$ phase at 850 °C revealed the existence of a metastable CIS phase ($\alpha_m$), shown by the dotted purple lines and patterned areas.

The reported existence of a metastable phase means that—from a thermodynamic point of view—there exists a region of relative stability (a relative minimum in the Gibbs free energy landscape) that may be reached via alternative sample histories, i.e., not necessarily by quenching.

The reported metastable CIS phase extends largely to the Cu-poor side by ca. 4 at.% Cu (red patterned area). Hence, maintaining single-phase CIS with Cu concentration as low as 18.5 at.% (Cu/In = 0.64) is possible. By comparison, highly

efficient CIGS is grown as a Cu-poor material with Cu/(In + Ga) ratios as low as 0.8 (21.6 at.% Cu)[17,20], i.e., potentially in a CIGS Cu-poor metastable homogeneity region. This should not be a surprise, because the deposition of CIGS thin films is a complex process performed out of thermodynamic equilibrium[21].

The extension of the metastable CIS phase towards Cu-rich compositions is narrower, but still appreciable (cyan patterned area), and clearly visible in the zoomed portion of the diagram (Fig. 1b). It may then be possible for Cu supersaturation to occur in CIS phases grown as thin films under large Cu excess and after unforced cooling (Fig. 1c).

The breadth of the phase homogeneity field at growth temperature implies a certain solubility of point defects such as indium on copper antisites ($In_{Cu}$) or copper vacancies ($V_{Cu}$) (or both) on the Cu-poor side, as well as copper on indium antisites ($Cu_{In}$), indium vacancies ($V_{In}$) or copper interstitial atoms ($Cu_i$) (or combinations of the three) on the Cu-rich side[22]. The solubility limit for these defects in the CIS lattice is even larger in the metastable phase.

The native doping density within this compositional window up to just above stoichiometry changes by nearly four orders of magnitude[23]. Both a drop of solubility limit from growth to room temperature (as proposed for sodium dopant[24]) and the possible occurrence of $\alpha_m$ are likely to have an impact on CIS processing (i.e., film microstructure during growth and reactivity after cooling), and hence PV performance.

**Composition-dependent CIS micro/nanostructure.** As revealed in the cross-sectional images (Fig. 1b), the microstructure of the films varies with increasing Cu content. Cu-poor CIS displays grains of maximum 1 μm size. Conversely, Cu-rich CIS has much larger grains, with at least double the lateral width and height up to 3 μm. The pronounced grain growth of Cu-rich films is generally attributed to the vapour–liquid–solid (VLS) mechanism exerted by Cu-Se fluxing phases during growth[25]. Crucially, stoichiometric CIS shows intermediate grain sizes, but the VLS mechanism cannot apply in this case, because no Cu-Se phases were formed during this 'single-stage' growth process.

Mainz et al.[26] have studied in detail and in-situ the annihilation of planar defects in Cu-poor CIGS during annealing with increasing Cu content. Importantly, they observed that the annihilation rate increases rapidly, shortly before $Cu_2Se$ phase segregates at the CIGS surface, and attributed domain growth to the resulting stress relaxation[27].

However, enhanced atomic diffusivity could also stem from copper supersaturation, especially in the presence of $Cu_i$ expected from the larger defects pool at growth temperature. $Cu_i$ defects could then migrate via the Frank–Turnbull dissociative mechanism, leading to lower migration barriers and faster atomic rearrangements[28].

The fact that the planar defect annihilation onset coincides with the incipient formation of $Cu_2Se$ is intriguing. To reveal the effect of growth conditions on the nanoscopic structure of CIS and test the existence of the supersaturated metastable phase, atom probe tomography (APT) is performed and shown in Fig. 2. APT analysis of stoichiometric CIS reveals a non-uniform distribution of Cu within the specimen. Some regions of the sample far exceed the stoichiometric Cu concentration, up to above 30 at.% near what appears to be a grain boundary (GB), as evidenced by alkali metal decoration[29], cf. region of interest 2. There is no clear-cut phase separation within the Cu-rich region to the left of the GB. The higher Cu content in the region is only weakly compensated by lower In and Se. This is compatible with the existence of metastable Cu supersaturation in single-stage CIS

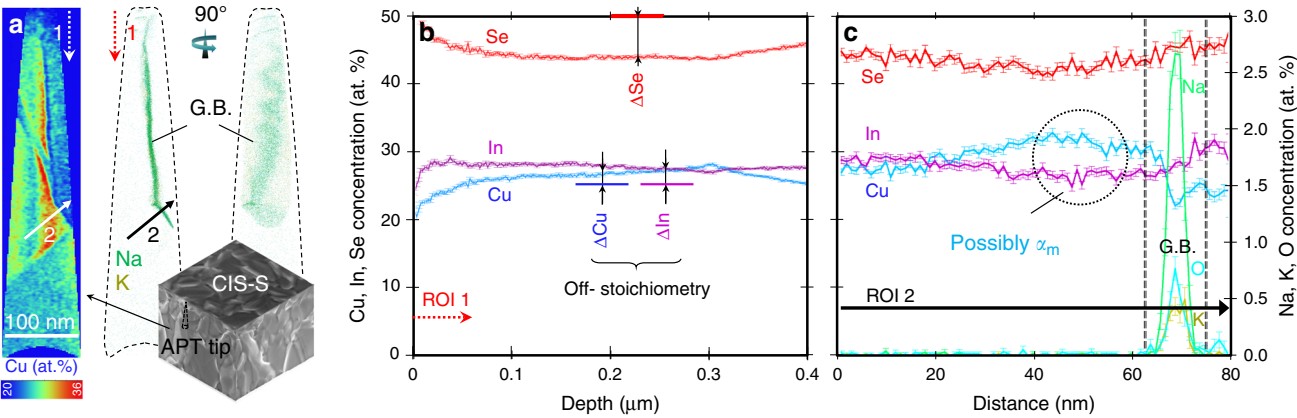

**Fig. 2 Atom probe tomography of stoichiometric CuInSe₂ (CIS-S). a** Cu concentration map (cf. Cu atomic % coloured scalebar) and corresponding Na and K elemental distributions, acquired to highlight the location of a grain boundary. The labelled arrows identify orientations of the regions of interests (ROI) 1 and 2. **b** Integral compositional depth profile from the whole specimen volume with highlighted off-stoichiometry: deficiency of selenium (ΔSe) and excess of copper (ΔCu) and indium (ΔIn) with respect to stoichiometric CIS (50 at.% Se and 25 at.% In and Cu). **c** Compositional profile along ROI 2, revealing a portion with Cu concentration far exceeding stoichiometry (dotted circle).

thin films, as reported by Gödecke et al.[8] for bulk material. Having detected such Cu-rich regions in stoichiometric CIS also suggests that Cu supersaturation may start to occur at nominal compositions corresponding to the incipient formation of Cu₂Se secondary phase (Cu-rich domains also occur in Cu-rich CIS, but are absent in Cu-poor CIS, cf. Supplementary Figs. 1 and 2).

One possibility (i) could be that the Cu-rich domains are accumulations of $Cu_i$ defects within the lattice of the metastable phase at room temperature, formed by focused ion beam (FIB)-induced aggregation and chemical reduction to elemental copper. Another possibility (ii) could be that the domains result from segregation of the excess $Cu_2Se$ at extended defects during cool down (Fig. 1c). Recent high-resolution TEM evidence from Simsek Sanli et al.[30] has revealed the presence of interspersed $Cu_2Se$ domains within CIGS grains, to which they attributed a topotactical grain growth effect during the three-stage process[31]. At this point, it is not possible to resolve with certainty which of the two scenarios occurs in reality.

Regardless of the actual mechanism of grain growth, the larger grains obtained under Cu-rich conditions display lower densities of detrimental planar defects[26], which is the reason why highly efficient CIGS is grown by deliberately crossing the edge of the homogeneity region during the so-called three-stage process[13].

Why then Cu-rich CIS consistently yields worse device performance than Cu-poor CIS, even after the removal of the excess $Cu_2Se$ phase[14] and despite the proven superior transport properties, is yet to be understood. Here, the composition-dependent reactivity of CIS towards atmospheric oxygen and post-deposition etchants is investigated to solve this conundrum in light of the APT compositional results on stoichiometric CIS and the possible existence of the supersaturated metastable $\alpha_m$ phase in CIS thin films.

**Thermochemistry and defect population metastability.** The thermodynamic stability of CIGS against reactions at interfaces has been assessed in finished devices[32–34] and on bare absorbers[35,36]. Kazmerski et al.[37] revealed the formation of $In_2O_3$, $SeO_2$ and $Cu_xSe$ on the CIS film surface above 150 ℃[38], with Hauschild et al.[36] and Lehmann et al.[39] finding $InO_x$ to be the most prominent oxide in air-exposed Cu-poor films. Figure 3a shows the computed Gibbs free energy of bulk $In_2O_3$ formation, respectively, out of $In_2Se_3$ [Eq. (1)] and CIS [Eq. (2)], as well as the alternative formation of $Cu_2O$ out of CIS [Eq. (3)].

$$In_2Se_{3(s)} + \frac{3}{2}O_{2(g)} \rightleftharpoons In_2O_{3(s)} + \frac{3}{n}Se_{n(s,l,g)} \quad (1)$$

$$2CuInSe_{2(s)} + \frac{3}{2}O_{2(g)} \rightleftharpoons In_2O_{3(s)} + Cu_2Se_{(s)} + \frac{3}{n}Se_{n(s,l,g)} \quad (2)$$

$$2CuInSe_{2(s)} + \frac{1}{2}O_{2(g)} \rightleftharpoons In_2Se_{3(s)} + Cu_2O_{(s)} + \frac{1}{n}Se_{n(s,l,g)} \quad (3)$$

Conversion of $In_2Se_3$ to $In_2O_3$ and elemental selenium formally corresponds to oxidation of selenide anions and is very favourable in air (negative $\Delta G$, solid red line). The positive slope of $\Delta G$ is consistent with the decreased degrees of freedom of the gas species, implying an even stronger thermodynamic driving force for conversion at room temperature.

Clearly, the Gibbs free energy of CIS formation is insufficient to stabilize the compound[34] (dashed red line); hence, unprotected CIS is intrinsically unstable against oxygen (cf. Supplementary Fig. 4 for detailed point defect reaction energetics). The thermodynamic driving force is a necessary but insufficient condition for a reaction to happen (cf. Supplementary Note 1). Nevertheless, Eq. (2) is confirmed to occur at room temperature (Supplementary Fig. 3) and—thermodynamically speaking—cannot be reverted even under exposure to a static atmosphere of metastable $Se_2$ molecules generated hypothetically by a cracker effusion cell (black dotted line). Stronger reagents, such as $H_2$, are needed to reduce $In_2O_3$[40].

Equation (3) leading to $Cu_2O$ is only barely favourable, so the formation of $In_2O_3$ is thermodynamically more likely, in agreement with Hauschild et al.[36] and Lehmann et al.[39].

The presence of humidity was shown to accelerate further the oxidation of the related GaSe 2D compound[41], so it is very likely that exposure of CIGS samples to (moist) air during processing affects the CIGS surface chemistry and the physics of the subsequent solar cell device.

To corroborate these considerations, a Cu-poor epitaxial CIS film, i.e., a proxy for CIS bulk (free from grain boundaries), was subject to a graded oxidation. The resulting library was characterized by scanning electron microscopy (SEM)/energy-dispersive X-ray spectroscopy (EDS) at low acceleration voltage and by photoluminescence (PL) along the macroscopic gradient, as shown in Fig. 3b. The regions closer to the O₂ source result in a

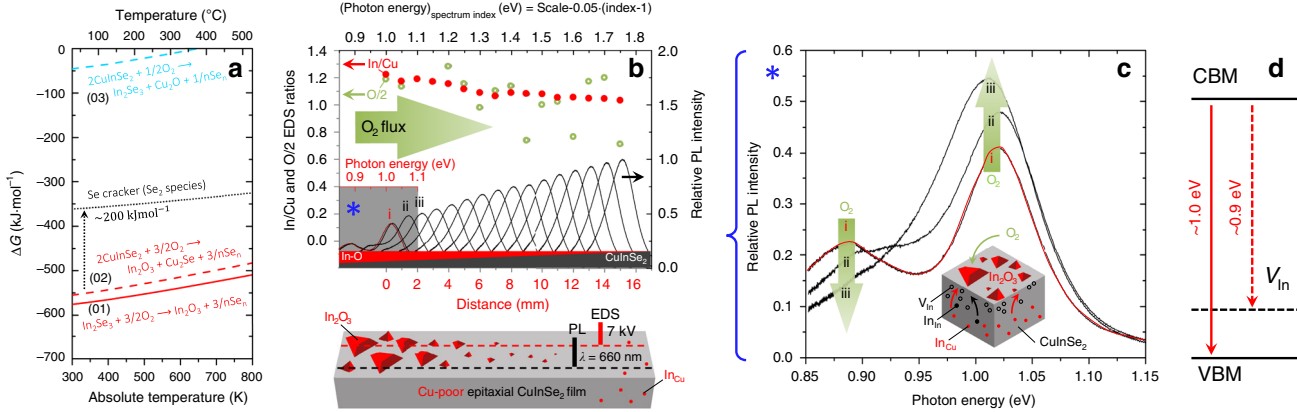

**Fig. 3 Thermochemical bulk stability of CuInSe$_2$ against oxygen. a** Gibbs free energy for the conversion of In$_2$Se$_3$ into In$_2$O$_3$ (solid red) and for the conversion of CIS into either In$_2$O$_3$ + Cu$_2$Se (dashed red) or In$_2$Se$_3$ + Cu$_2$O (dashed cyan). The dotted black line corresponds to the energetic offset induced by a selenium cracker hindering (but not reverting) In$_2$O$_3$ formation. **b** Combinatorial room-temperature PL (right ordinate) and EDS (left ordinate) analysis of an epitaxial Cu-poor CIS film annealed at 550 °C in the presence of an O$_2$ flux, leading to a library sample with graded In$_2$O$_3$ outgrowth on the CIS surface, as shown in the schematics. **c** Room-temperature PL spectra of the three leftmost interrogated areas of the library sample in **b**, i.e., of the area that incurred the largest extent of oxidation, under the same photon energy axis. **d** Energy diagram of the electronic transitions occurring in the epitaxial Cu-poor CIS film as a result of deliberate oxidation (cf. ref. [43]). The growth of In$_2$O$_3$ thermodynamically proceeds first by consumption of the least stable atomic indium species (In$_{Cu}$ antisites) near the surface, followed by the more stable ones (In$_{In}$), leading to the formation of indium—and copper—vacancies (V$_{In}$), as shown by the schematic illustration.

higher In/Cu ratio (full red dots) and O content (hollow dots), consistent with the outgrowth of an In-O phase. Concomitantly, the PL yield (black curves) is lower and a shoulder peak appears, as shown more clearly in Fig. 3c.

To gain mechanistic insights into the CIS surface reactions, it is useful to relate the thermochemistry at macroscopic level to the physics and chemistry of point defects in the material[42]. The shoulder peak at ca. 0.9 eV is ascribed to the transition from the conduction band minimum to the indium vacancy (V$_{In}$) in CIS[43]. It is hypothesized that the formation energy of V$_{In}$ in CIS (ca. 2.5 eV/V$_{In}$[44,45]) resulting from the reaction between indium atoms and gas-phase oxygen is offset energetically by the large Gibbs free energy gain associated to Eq. (2) (ca. 2.9 eV/In atom).

The formation of V$_{In}$ under kinetic control implies that the outgrowth of In-O does not occur entirely as per Eq. (2), but leads —at least partially—to a defected CIS lattice without the segregation of Cu-Se phases, as per Eq. (4).

$$2[Cu_{(1-y-z)}(In_{Cu})_y(V_{Cu})_z]InSe_{2(s)} + \frac{3x}{2}O_{2(g)} \rightarrow$$
$$xIn_2O_{3(s)} + 2[Cu_{(1-y-z)}(In_{Cu})_y(V_{Cu})_z][(In)_{1-x}(V_{In})_x]Se_{2(s)} \tag{4}$$

From the phase diagram viewpoint, Eq. (4) is ensured by the Cu solubility range (Fig. 1b), entailing an increase of the Cu/In ratio of the underlying Cu-poor CIS without Cu$_2$Se ejection. Under thermodynamic equilibrium, the increased Cu/In eventually translates into a lower share of (In$_{Cu}$ + V$_{Cu}$), but from a kinetic standpoint V$_{In}$ defect formation is justified: given the relatively high energy barrier for In migration in CIS (ca. 1.1 eV[46]), once V$_{In}$ is formed it is not easily dissolved.

It is then revealed that interface reactions can lead to a chalcopyrite material with metastable defect populations at temperatures as high as 550 °C.

### Composition-dependent CIS reactivity against O$_2$.
Equation (2) is now assessed at room temperature as a function of the Cu content in CIS. Figure 4a shows the X-ray photoemission spectroscopy (XPS) signature of Cu-poor, stoichiometric and Cu-rich CIS after air exposure of the single-phase films for 24 h (i.e., films

previously subject to removal of the excess Cu$_2$Se by surface chemical etching with potassium cyanide (KCN)).

The surface composition of air-exposed Cu-poor CIS is consistent with In-O outgrowth, at least partially compatible with Eq. (4), although Cu-Se segregation is not excluded considering that atomic redistribution at room temperature is slower than at 550 °C. Stoichiometric CIS shows a similar behaviour.

Conversely, the surface composition of air-exposed Cu-rich CIS reveals a higher Cu-Se concentration, which is compatible with larger Cu-Se outgrowth. A reaction pathway alternative to Eq. (4) should then be invoked to account for the XPS data in this case. If Cu-rich CIS is a Cu supersaturated $\alpha_m$ solution, the Cu-Se outgrowth may proceed by consumption of Cu interstitial atoms and formation of V$_{Se}$, as per Eq. (5), where indium oxide formation leads to In subtraction from a material already oversaturated with Cu.

This can exclusively be compensated by the formation of the Cu-Se secondary phase in Cu-rich material, unless the limit of Cu equilibrium solubility in CIS is exceeded.

$$2[Cu(Cu_i)_z][In_{(1-z)}Se_{(1-z/2)}]_{(s)} + \frac{3x}{2}O_{2(g)}$$
$$\rightarrow xIn_2O_{3(s)} + yCu_2Se_{(s)}$$
$$+ 2[Cu(Cu_i)_{(z-y)}][In_{(1-z-x)}(V_{In})_x][Se_{(1-z/2-y/4)}(V_{Se})_{y/4}]_{2(s)} \tag{5}$$

Figure 4b, c exemplify the extreme cases of interface reactions from the perspective of CIS point defects under Cu deficiency and Cu excess, as inferred from the experimental evidence. The formation of indium oxide provides most of the thermodynamic driving force for Eq. (5) to happen, but the Cu excess in Cu-rich CIS appears to increase the reaction rate compared with Cu-poor CIS, which can be represented by a lowering of the reaction energy barrier, as suggested in Fig. 4d.

It is proposed that the nucleation and growth of the Cu$_2$Se phase in the Cu-rich film is eased either by the large pool of Cu interstitials present in the Cu supersaturated $\alpha_m$ solution and/or by the pre-existence of Cu$_2$Se domains acting as seeds.

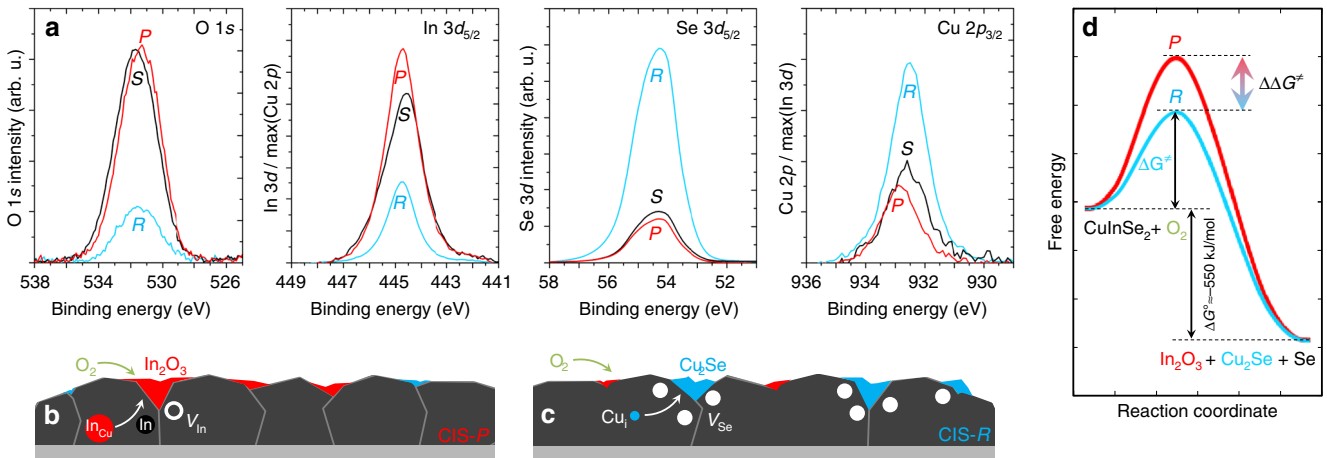

**Fig. 4 Composition-dependent CuInSe₂ surface reactivity against O₂ at room temperature. a** XPS surface analysis of Cu-poor CIS (P, red), stoichiometric CIS (S, black) and Cu-rich CIS (R, cyan) films after removal of Cu-Se phases by KCN (bold)etching and subsequent exposure to air at room temperature for 24 h. Schematics representation of point defect oxidation reactions in CIS-P (**b**) and CIS-R (**c**) drawn to match the XPS results, i.e., larger share of In-O outgrowth in Cu-poor material and of Cu-Se in Cu-rich material. **d** Diagram of free energy vs. reaction coordinate for the oxidation of CuInSe₂ to In₂O₃, Cu₂Se and Se showing the spontaneous (negative) bulk Gibbs free energy of reaction at room temperature ($\Delta G°$), and the proposed occurrence of different Gibbs free activation energies ($\Delta G^{\neq}$) for the oxidation pathways under Cu-poor and Cu-rich CIS compositions.

It is then established that interface reactions can have profound consequences for the underlying defect chemistry of chalcopyrite even at room temperature without any liquid phase being involved. Furthermore, the defect reaction pathways can be markedly diverse if the substrate phase happens to occur near the edges of its existence region, which can have fundamental implications for further processing and ultimately for device performance.

**Composition-dependent CIS stability against cyanide.** Cu-Se phases are very conductive and reduce severely the shunt resistance of copper-based chalcogenide solar cells[18,25]. Whether they occur as residues of absorber fabrication or following exposure to ambient air, they are effectively removed by aqueous cyanide (CN⁻) solutions, presumably following Eq. (6) or Eq. (7).

$$Cu_2Se_{(s)} + 7KCN_{(aq)} + \frac{1}{2}O_{2(g)} + H_2O_{(l)}$$
$$\rightarrow 2K_2\left[Cu(CN)_3\right]_{(aq)} + KSeCN_{(aq)} + 2KOH_{(aq)} \quad (6)$$

$$Cu_2Se_{(s)} + 6KCN_{(aq)} \rightarrow 2K_2\left[Cu(CN)_3\right] + K_2Se_{(aq)} \quad (7)$$

KCN etching is also known to restore, at least partially, the original optoelectronic properties of aged absorber layers[47,48], even after oxidation by strong agents such as hypochlorite[19]; hence, PV device fabrication baselines may include a cyanide etching step, enabling a diode-like electrical behaviour[18,49]. However, unlike widely accepted, cyanide is unable to chemically remove the In₂O₃ phase formed as a result of CIS oxidation, i.e., any removal of such a phase upon cyanide etching is consequential to the chemical dissolution of interspersed Cu-Se phases, as per Eq. (6) or Eq. (7) (cf. Supplementary Fig. 3).

Having established the composition-dependent reactivity of CIS towards ambient air, it is important to understand whether the classical KCN etching also alters the defect population near the interface[50] differently, depending on the CIS bulk composition and etching conditions. To this end, unetched films (KCN-U) are compared with films subject to two etching conditions: a 'weak' (KCN-W) etch (5 wt.% KCN for 30 s duration) and a 'bold' (KCN-B) etch (10 wt.% KCN for 300 s duration) on selected samples of Cu-poor, stoichiometric and Cu-rich CIS. The sample

naming in the following figures is attributed combining the CIS chemical composition ('Poor', 'Stoichiometric' or 'Rich') with the KCN etching condition ('Unetched', 'Weak etch' or 'Bold etch'), e.g., yielding PW for Cu-poor with weak KCN etch (cf. Table 1 for full naming convention).

SEM/EDS analysis shows the morphological and compositional effects of KCN etching near the CIS surface and deeper in the films (Fig. 5a, b). Figure 5a is the corresponding matrix of top views. The small clusters identified at the surface of Cu-poor unetched and Cu-rich weakly etched CIS films are attributed to remnant Cu₂Se phase, as they are clearly removed with further etching. EDS yields compositional information that is reasonably free from roughness convolution. At 7 kV acceleration voltage, the beam interacts approximately with the topmost 200 nm of film, whereas at 20 kV the interaction reaches at least 800 nm depth. It follows that Cu-rich CIS after bold etch shows a more pronounced surface Cu depletion than Cu-poor after bold etch, which is striking considering that the bulk composition is by far the opposite (consistent with the growth conditions and with APT analysis in Supplementary Fig. 1).

The Se content is described by the (Cu + In)/Se ratio. Cu-poor CIS films display values around unity in the bulk and slightly above unity at the surface, irrespective of the KCN etching conditions. Conversely, the Se content of Cu-rich CIS films depends heavily on the type of KCN etching because of the large amounts of Cu₂Se in Cu-rich unetched and their partial removal by weak KCN etching. However, importantly, the value of (Cu + In)/Se at the surface of Cu-rich CIS after bold etch is above unity, indicative of surface Se deficiency, given that this film is surely free from Cu₂Se. The higher (Cu + In)/Se ratio in the bulk of Cu-rich CIS after bold etch may seem puzzling; however, it is explained by the presence of the Cu-rich domains identified by APT, in line with the higher Cu/In ratio at 20 kV.

Similar conclusions are drawn by means of secondary ion mass spectrometry (SIMS) depth profiling (Fig. 5c). However, when comparing SIMS with EDS, it should be borne in mind that SIMS is affected by film roughness, due to signal averaging (cf. Fig. 5a), and matrix effects. For example, the convolution of topography and matrix effects influences heavily the Cu and Se SIMS yields (hence, the SIMS profiles) of Cu-rich CIS after weak etch, where remnant Cu₂Se occurs.

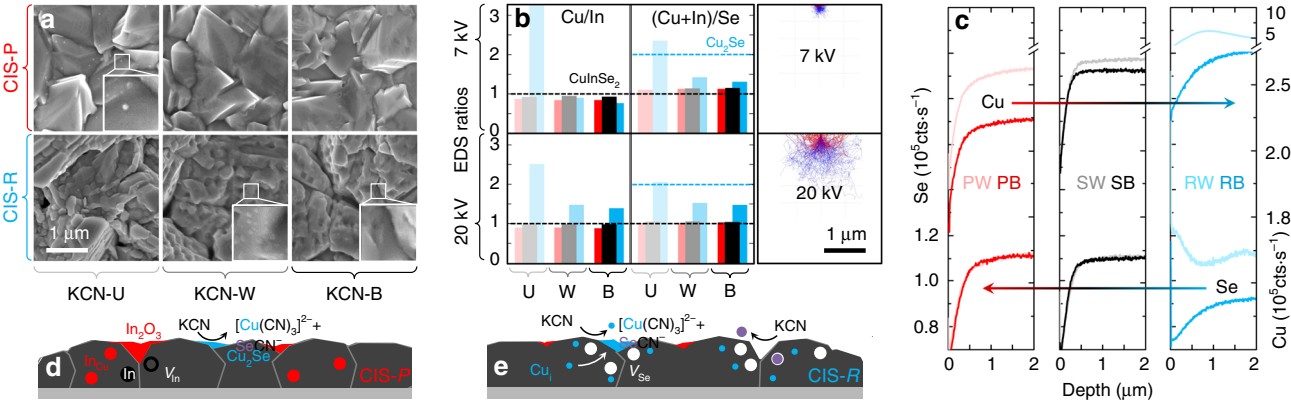

**Fig. 5 Composition-dependent CuInSe$_2$ surface reactivity against KCN at room temperature. a** SEM top views of Cu-poor CIS (CIS-P, top row) and Cu-rich CIS (CIS-R, bottom row) films unetched (KCN-U) and after KCN weak (KCN-W) and bold (KCN-B) etchings. The white squares are magnifications of selected areas to reveal the presence of Cu$_2$Se particles in PU and RW and their absence in RB. **b** Box plot of Cu/In and (Cu+In)/Se atomic concentration ratios acquired by EDS at 7 kV (top) and 20 kV (bottom) acceleration voltages, entailing beam-film interactions as shown by the corresponding CASINO[73] depth simulations. The dashed lines correspond to the ratios expected from pure stoichiometric CuInSe$_2$ (grey) and Cu$_2$Se (cyan). It is noteworthy that the Cu/In ratio of CIS-R exceeds 1 even at 20 kV, because the sample was intentionally grown under very high Cu flux, whereas the initial Cu/In ratio of standard Cu-rich films never exceed 1.3. **c** Cu and Se SIMS depth profiles of CIS-P (red), CIS-S (black) and CIS-R (blue) films subject to KCN-W (light shades) and KCN-B (dark shades). Cu and Se appear to be leached mostly out of CIS-R, whereas CIS-S shows the least compositional deviation. The Se profiles of CIS-P and CIS-S are virtually unaffected by KCN etching. **d, e** Schematic representations of point defect-driven selective leaching by KCN.

Both EDS and SIMS show that stoichiometric CIS yields the smallest compositional change upon KCN etching.

Overall, it is hypothesized that Cu atoms are removed selectively from the lattice of CIS by KCN; the more so, the higher the deviation from stoichiometry. Furthermore, Se depletion is apparent in Cu-rich CIS, leading to the formation of both Se and Cu vacancies. The extent of removal by the cyanide complexant is thermodynamically more favourable for Cu-rich than for Cu-poor CIS, possibly due to the abundance of Cu interstitial atoms in the supersaturated metastable phase, in line with the argument described in Fig. 4d (cf. Supplementary Fig. 4).

**Optoelectronics of CIS metastability induced by cyanide.** Having shown that KCN etching induces different chemical effects at the surface of CIS depending on the growth conditions, the next step is to investigate the effects of such surface alterations on the optoelectronic properties. To this end, photoelectrochemical (PEC), time-resolved surface photovoltage (TR-SPV), PL and thermal admittance (ADM) analyses were performed, giving complementary insights on the most relevant samples.

It is crucial to understand that the time constants discussed for PEC and SPV analyses are not carrier lifetimes, but time constants of electron transfer and of defect transformation/generation. A PEC current or photocurrent ($J_{Ph}$) is defined as the difference between the electrochemical current recorded under illumination and under dark conditions. Negative values, such as those observed here, correspond to photogenerated electron collection by the redox couple in solution from $p$-type CIS (Supplementary Fig. 5). The effect of bold KCN etching on the PEC transients is shown in Fig. 6a. Cu-poor CIS after bold etch (red) shows the typical photocurrent decrease with time after light perturbation until steady state. The behaviour follows the theory of 'surface recombination' at semiconductor/electrolyte junctions, treated extensively by Peter[51]. Here, it is stressed that the term 'recombination' used in this context does not refer to the typical fast decay discussed in PL or cathodoluminescence, due to electron-hole recombination, but to electron transfer to the solution and to the change of charge state of surface defects. The time constants are not carrier lifetimes, but the times needed for the electron transfer or for the (re-)charging of the interface

defects. Recombination at the interface (or surface recombination, s.r.) is negligible upon light perturbation if the photogenerated carriers are collected fast enough by the redox couple (i.e., with a high kinetic constant for electron transfer, $k_{tr}$). However, if the electron transfer is sufficiently slow ($k_{tr}$ lower than $k_{s.r.}$), a bottleneck is created, i.e., the minority carriers accumulate near the interface over time, leading to recombination through leaky surface states and a decrease of photocurrent until steady state. This recombination pathway effected by surface states is most probably due to electrically conductive Cu-Se phases, as no quasi-Fermi level splitting can exist in metal-like solids. The inset compares the $k_{s.r.}$ of Cu-poor and Cu-rich CIS, both after weak etch (both of which display this typical transient) determined as a function of air-exposure time after etching at room temperature [cf. Eq. (8) in "Methods" section]. After weak etching, Cu-rich shows higher $k_{s.r.}$ than Cu-poor CIS and a higher rate of increase, compatible with the larger amount of Cu-Se at the surface (incomplete removal by weak KCN etching, Fig. 5a) and surface oxidation behaviour (XPS, Fig. 4).

The shape of the photocurrent transient in Cu-rich CIS after bold etch is very unusual, because $J_{Ph}$ increases with time after the start of the light perturbation (note that the figure plots the measured current density that comprises dark current and photocurrent). Evidently, a different additional recombination or trapping pathway induced by the bold KCN etching must be invoked to account for this uncommon behaviour. The situation is explained by the presence of another defect(s) inside the absorber near the surface with different time constant (tentative band diagram in Fig. 6a), which saturates slowly. The observed time constant is not the carrier lifetime, but the time needed to change the charge state of the involved defects(s).

After bold etch, the stoichiometric CIS film shows a PEC transient under illumination that is a linear combination between those of Cu-poor and Cu-rich subject to the same bold etch. This confirms undoubtedly that the concentration of the point defect(s) formed upon bold KCN etch depends heavily on the Cu/In composition, hence the Cu and In chemical potentials in CIS, with Cu-rich CIS being the most susceptible to the defect(s) formation.

Intensity-modulated photocurrent spectroscopy (IMPS) was performed to gain quantitative information on the slow recombination

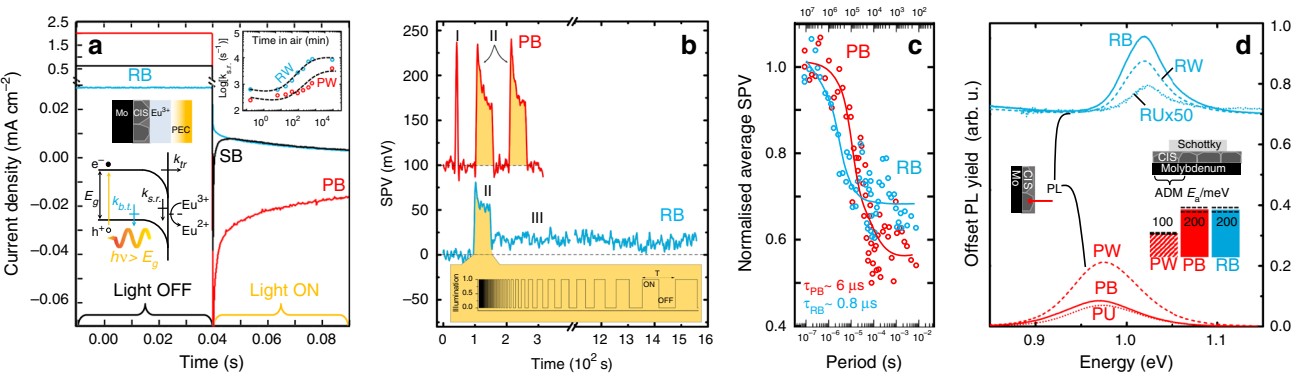

**Fig. 6 Electrical identification of the metastable defects induced by cyanide etching on CuInSe₂ absorbers. a** PEC transients of Cu-poor CIS (P, red), stoichiometric CIS (S, black) and Cu-rich CIS (R, blue) films immediately after bold KCN etching (**b**) (hence: PB, SB and RB). Inset: evolution of the surface recombination kinetic constant ($k_{s.r.}$) of CIS-P (red) and CIS-R (blue) as a function of air-exposure time after weak KCN etching (W) (hence, PW and RW). Schematics: band diagram of the CIS/electrolyte junction with an additional recombination pathway and corresponding bulk trap kinetic constant ($k_{b.t.}$). **b** TR-SPV results on CIS- P (red, shifted by 100 mV) and CIS-R (blue) with bold KCN etching (B). The relaxation curve of CIS-PB acquired in real time after excitation with a continuous-wave light source with 1–2 s duration (Region I) shows a fast decay time, which cannot be resolved by KPFM. To resolve it, modulated illumination was used to obtain frequency spectra (during which a partial SPV decay is observed, region II). Unlike CIS-PB, where the SPV drops sharply to the initial in-dark value after the modulation sequence, CIS-RB does not recover completely (region III). The light modulation sequence has a 50% duty cycle for a total duration of 5.2 s, as shown by the schematic square-wave in the yellow inset. **c** Normalized average SPV spectra as a function of the illumination modulation frequency (corresponding to region II). Decay times ($\tau$) are determined by fitting (solid thick lines) the data (open circles) with an exponential behaviour yielding: $\tau_{PB} \sim (6.2 \pm 1.0)$ μs and $\tau_{RB} \sim (0.75 \pm 0.2)$ μs (cf. "Methods"). **d** PL yield of CIS-P (red) and CIS-R (blue) films without etching (U, dotted), and after weak KCN etching (W, dashed) and bold KCN etching (B, solid line). Inset: activation energies of the main capacitance step extracted from the slope of the Arrhenius plots of the thermal admittance spectra (320–50 K) of Schottky junctions with CIS-P (red) and CIS-R (blue). Bold KCN etching yields an activation energy of 200 ± 20 meV on both CIS-P and CIS-R, whereas CIS-PW shows an activation energy of 100 ± 10 meV.

pathway apparent in Cu-rich CIS after bold etch (Supplementary Fig. 6). In fact, after bold etch, both Cu-rich and Cu-poor films display recombination pathways with long time constants (2.2 μs and 2.5 μs, respectively). However, the extent of recombination for Cu-rich is ca. one order of magnitude higher than for the Cu-poor case, such that the pathway is perceived in the time domain of the Cu-rich film, but not in that of the Cu-poor one.

TR-SPV measured by Kelvin probe force microscopy (KPFM) is used to analyse the effect of bold etching on the photogenerated carrier dynamics of Cu-poor and Cu-rich films (Fig. 6b, c). The SPV, defined as the difference between the contact potential difference (CPD) under illumination and the CPD in the dark, is measured as a function of the modulation frequency of a light source. Regions II of the SPV (time domain) transients show an initial fast response reaching a maximum SPV at the beginning of the illumination sequence (starting at 10 MHz), followed by a slow decay and an abrupt drop of the SPV signal at the end of the sequence (finishing at 100 Hz). Upon initial illumination, both samples show a positive SPV, consistent with a p-type material behaviour. However, Cu-rich CIS after bold etch yields an SPV of ∼75 mV, which is approximately half the value obtained for the analogous Cu-poor film. A lower SPV is indicative of a smaller change of band bending at the surface, which can be related to a lower minority charge accumulation at the illuminated surface of CIS-RB. The average SPV spectra in the frequency domain (Fig. 6c) reveal decay times of $\tau_{CIS-PB} \sim 6.2 \pm 1.0$ μs and $\tau_{CIS-RB} \sim 0.75 \pm 0.2$ μs. This trend is consistent with a higher recombination rate for Cu-rich CIS after bold etch, in agreement with the relative $k_{s.r.}$ of Cu-rich and Cu-poor films subject to bold etch obtained from PEC.

During the illumination sequence, the average SPV of Cu-poor CIS after bold etch decays to ca. 50% of the initial value, indicating a full relaxation of the SPV (for sufficiently slow modulation above 10 kHz, the SPV is at the maximum and at 0 for half of the cycle, giving an average of 50%). In fact, when the light excitation is off after the modulation spectrum, the CPD

value in dark conditions is fully recovered. However, the CPD of Cu-rich CIS after bold etch does not drop to the dark value after finishing the modulation spectrum, but a finite SPV value is maintained at about 20 % of the total SPV for at least ca. 26 min after switching the excitation light off (Region III). Thus, the average SPV decays only by 30% from the SPV maximum, indicating that the fast decay of the SPV does not lead to a full relaxation. The remnant SPV is attributed to the presence of charged trap states. This finding is consistent with the presence of additional recombination or trapping pathways in Cu-rich CIS after bold etch already identified by PEC.

The PL spectra (Fig. 6d) are acquired immediately after the etching (less than 3 min), to minimize surface degradation[48]. For Cu-poor CIS, the highest PL yield is displayed by the film subject to weak etch, showing a fourfold increase compared with the unetched, consistent with the known CIS surface renewal by cyanide etching with the removal of Cu₂Se (Fig. 5a). However, bold etching appears to be detrimental for PL yield, suggesting the formation of detrimental defects.

For Cu-rich CIS films, bold etching leads to higher luminescence than weak etching. This is attributed to the fact that weak KCN etching is unable to remove the entire Cu-Se phases (cf. Fig. 5a) responsible for reduced photon absorption by CIS and subsequent emission due to light scattering, as it is most evident for the unetched film.

The energy of the PL maxima of Cu-poor and Cu-rich CIS films reflects the different bandgaps of the absorbers[52] and the different extent of electrostatic potential fluctuations[53].

The bar chart in Fig. 6d shows the relative activation energy ($E_a$) of the main capacitance step extracted from the slope of the Arrhenius plot of the ADM spectra of Schottky junctions formed with the bare absorbers (Supplementary Fig. 7). The activation energies provide information on the electrically active defects in the material. A value of 200 ± 20 meV is obtained if bold KCN etching is performed, regardless of the CIS composition and cyanide counterion (cf. Supplementary Fig. 7), twice as much as for Cu-

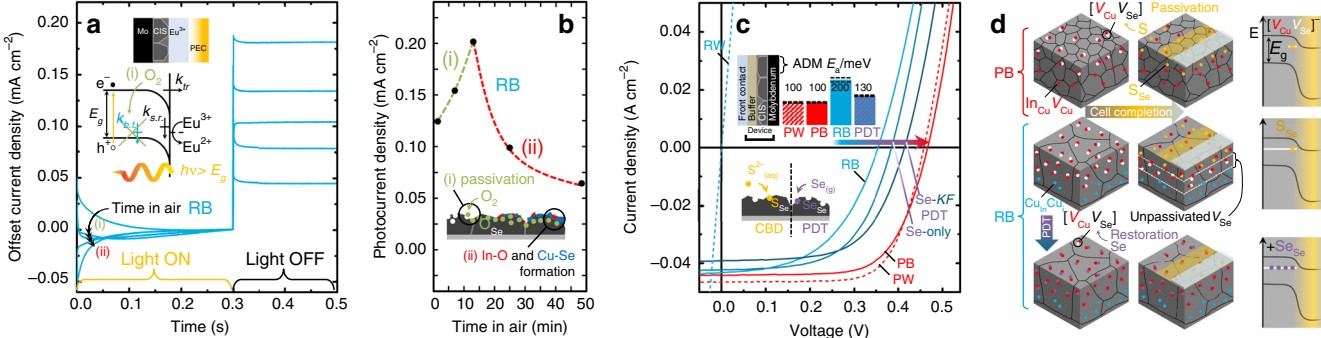

**Fig. 7 Chemical identification/passivation of the CuInSe₂ metastable defects and impact on PV device performance. a** PEC transients of Cu-rich CIS subject to bold KCN etch (RB) after repeated exposure to air at room temperature (note that illumination starts at time 0 and, for discussion purposes, the curves have been y-shifted such that the steady-state current densities under illumination coincide). **b** Evolution of the photocurrent density $J_{Ph}$ as a function of air-exposure time, revealing an initial regime of photocurrent increase (i) exemplified in the schematics by O₂-induced passivation of $V_{Se}$ donors and corresponding removal of the bulk trap recombination pathway, followed by a subsequent decrease (ii), due to the detrimental formation of Cu-Se an In-O at the CIS surface. **c** JV characteristics of solar cell devices comprising Cu-poor (P, red) and Cu-rich (R, blue) CIS absorbers subject to weak KCN (W, dashed) and bold KCN (B, solid) etching. The Voc improvement obtained with CIS-RB films upon Se-only and Se-KF PDTs is shown by the arrow. The bar chart refers to the corresponding activation energies of the main capacitance step extracted from thermal admittance spectroscopy (see Supplementary Figs. 7 and 8). **d** Proposed schematic illustration of point defect distribution in Cu-poor and Cu-rich films after bold KCN etching (PB and RB), showing the different extent of Cu and Se depletion leading to the formation of $[V_{Cu}V_{Se}]$ divacancy complexes. The divacancy complex occurs near the surface of CIS-PB but extends deeper in the bulk of CIS-RB. The deposition of CdS by chemical bath close to room temperature (solar cell completion) induces S incorporation into $V_{Se}$, i.e., dissolution of the complex. The S incorporation is sufficient to passivate the corresponding electronic states present at the surface of PB film but is insufficient to reach deeper in the RB film; hence, some of the defect complexes remain. Post-deposition treatments with Se at high temperature prior to cell completion enable a more effective replenishment of the Se vacancies, with a beneficial passivating effect that increases the $V_{OC}$ of the photovoltaic cell.

poor after weak etching (100 ± 10 meV). No spectra could be recorded on unetched films and Cu-rich films after weak etching, due to the presence of the conductive Cu-Se phases at the surface.

A recent DFT study by Simsek Sanli et al.[54] on the effect of $Cu_{In}$ antisites and Cu interstitials in CIS reveals the appearance of states at 270 meV above the valence band maximum and 210 meV below the conduction band minimum, respectively. However, it is not obvious why the population of these two defects would increase upon KCN etching. Instead, the 200 ± 20 meV defect could be related to the observed etching-induced depletion of Cu and Se from CIS. Besides, the $Cu_{In}$ and $Cu_i$ have clear PL signatures (cf. Spindler et al.[4]) and do not correlate with the 200 ± 20 meV defect.

**Passivation of metastable defects improves PV efficiency.** To shed light on the Cu and Se depletion, and link it to the trap defect (s) identified by PEC and ADM, the surface chemical reactivity of Cu-rich CIS after bold etch is investigated as a function of air-exposure time (Fig. 7a, b). It is revealed that oxygen has a remarkable influence on the shape of its anomalous PEC transient (blue curves shown under illumination) and magnitude of photocurrent (inset). The recombination pathway induced by bold KCN etching appears to be slowly passivated by oxygen incorporation, leading to a transition from the anomalous type of transient observed for Cu-rich (Fig. 6a, cyan) to the classical transient typical of Cu-poor films (Fig. 6a, red) and simultaneous increase of photocurrent density ($J_{Ph}$, inset). The photocurrent density reaches a maximum after 12 min of air exposure and then it decreases, due to interface deterioration caused by the out-growing Cu-Se and In-O surface phases, as per Eq. (5).

Whatever the nature of the defect created by the bold KCN etching, oxygen appears to neutralize it, reducing the near-surface recombination pathway. It is plausible that oxygen passivates selenium vacancy donor defects, as proposed by Kronik et al.[55]. The Se vacancies formed during etching would effectively compensate or even type-invert the near-surface region of CIS,

causing efficient capture of the photogenerated electrons, as observed within the PEC transient ($k_{tr}$ lower than $k_{b.t.} \sim k_{s.r.}$). During subsequent air exposure, the vacancies would capture oxygen, forming compensating acceptors that remove the electron trap and restore p-type doping.

To provide a hint on the nature of such a defect and its effect on the performance of solar cells, the composition-dependent performance of CIS solar cells is now investigated with respect to the chemical reactivity towards surface treatments. Solar cells comprising Cu-poor and Cu-rich absorbers subject to either weak or bold etching were fabricated following a standard baseline including CdS layer deposition by chemical bath (CBD). The electrical characteristics of the devices under simulated sunlight are shown in Fig. 7c.

The different etching does not affect appreciably the performance of Cu-poor-based devices in terms of electrical characteristics (red curves in Fig. 7c). Conversely, no valuable electrical data could be extracted from devices based on Cu-rich unetched and weakly etched absorbers, due to the residual Cu₂Se and resulting resistor-like behaviour (cf. RW dashed cyan curve). Only with bold KCN etch, it is possible to effect a diode curve out of an absorber with Cu-rich CIS composition (RB solid cyan curve).

However, it is clear that the device based on Cu-rich CIS subject to bold KCN etching is affected by a significant deficit of open circuit voltage ($V_{OC}$) compared with devices based on Cu-poor absorbers, regardless of the KCN etching type. In light of the PEC and ADM Schottky analysis, bold KCN etching could then be responsible for the formation of defects that cause recombination at the interface between the surface of Cu-rich CIS after bold etch and the buffer layer, hence the enduring $V_{OC}$ deficit of Cu-rich compared with Cu-poor CIS-based devices[56]. A series of experiments was performed to test this hypothesis.

If the origin of the $V_{OC}$ deficit is linked to the presence of Se vacancies, then post-deposition treatments (PDTs) performed on the absorber surface under Se-rich environments, should improve the $V_{OC}$. Three types of such PDTs were performed: Se-KF (selenium and potassium fluoride), Se-only and Se-In[57,58]

(cf. Table 1 and 'Methods' section). All three treatments effectively improve the $V_{OC}$ (cf. JV curves in Fig. 7c and ref. [57]), strengthening the hypothesis that the detrimental defect introduced by bold KCN etching is indeed related to Se depletion (the changes observed after the PDTs are linked to the Se incorporation and are not merely due to the associated heating).

To link the solar cell performance to the electronic defects identified by PEC and ADM on the bare absorbers, the PV devices were also characterized by ADM. The activation energies extracted from the Arrhenius plots are shown by the bar chart in Fig. 7c.

ADM reveals that the $200 \pm 20$ meV defect of Cu-poor and stoichiometric CIS films after bold etching disappears after the absorbers are subject to device completion (Supplementary Fig. 7), consistent with the fact that the etching type has negligible effects on the performance of Cu-poor-based devices. The activation energies become $100 \pm 10$ and $130 \pm 10$ meV, similar to those of devices based on weakly etched absorbers with Cu-poor and stoichiometric composition, which are close to the common A2 (100 meV) and A3 (130 meV) acceptor defects found in PL[43,59]. The removal of the $200 \pm 20$ meV defect in the devices based on Cu-poor and stoichiometric absorbers is consistent with the passivation of Se vacancies via sulfur incorporation during the baseline deposition of the CdS buffer layer by CBD.

Conversely, along with the significant $V_{OC}$ deficit, devices based on Cu-rich absorbers subject to bold etch also retain the $200 \pm 20$ meV defect, unlike cells based on Cu-poor CIS. It is important to note that both this defect and the $V_{OC}$ deficit are observed in all devices based on Cu-rich absorbers subject to bold etch regardless of the Cu/In ratio (above 1) or Se flux during absorber growth[57]. Crucially, Se-based PDTs succeed in passivating the $200 \pm 20$ meV defect, uncovering the $130 \pm 10$ meV A3 acceptor[60] and improving $V_{OC}$. Therefore, a link is established between the $V_{OC}$ deficit, the $200 \pm 20$ meV defect and the bold KCN etching on Cu-rich CIS absorbers.

Moreover, devices comprising Cu-rich CIS absorbers subject to bold KCN etching and Zn(O,S) layers deposited by CBD with eight times the concentration of sulfur source compared with CdS CBD do not display the $200 \pm 20$ meV defect and also show a significant $V_{OC}$ improvement compared with standard CdS-based devices comprising Cu-rich films subject to bold KCN etching (Table 2). It is then clear that the longstanding $200 \pm 20$ meV defect is responsible for the $V_{OC}$ deficit of devices based on Cu-rich CIS subject to bold etch due to the preferential formation of Se vacancies induced by bold KCN etching on the metastable Cu-rich CIS phase. If Cu supersaturation of this phase is in the form of interstitial Cu atoms (cf. APT in Fig. 2), the coordination of Cu

and Se by the cyanide ions in solution would be thermodynamically more favourable compared with Cu-poor CIS.

The hypothesis is consistent with the near-surface composition revealed by EDS and with positron annihilation studies by Uedono et al.[61]. Cu and Se vacancies can cluster to form Cu-Se divacancy defect complexes. These complexes can account for larger losses in open circuit voltage than mere Se vacancies. Indeed, the energy level of the divacancy complex occurs deeper in the gap than the Se vacancy alone[62]. It follows that the improved performance upon the various PDTs is in accordance with the replenishment of Se vacancies with Se, even if the Cu vacancy population remains unchanged. Based on these considerations, Fig. 7d shows a conclusive schematic illustration of the proposed metastable defect population in CIS subject to bold KCN etching and of the passivating effects of CdS CBD and Se-PDTs.

The impact of the findings on solar cell performance is summarized in Table 2, where the device parameters of the various samples are reported.

**Metastability is caused by amphoteric ($V_{Se}V_{Cu}$) divacancies.** The theoretical model published by Lany and Zunger in 2006[62] provides a final support to the hypothesis that the preferential Se and Cu depletion caused by cyanide etching is responsible for the appearance of the defect identified by ADM with an activation energy of $200 \pm 20$ meV.

Figure 8 illustrates a simplified model consistent with the experimental evidence gathered by PEC and TR-SPV (cf. Fig. 6), describing the sequence of events occurring from light perturbation to relaxation. Given that the 200 meV defect is inaccessible by PL due to current detector limitations, the combined information from all the experimental techniques employed in this study and first-principle analysis sheds light on the chemical nature of this yet unidentified defect, i.e., complementing the recent review on CIGS electronic defects[4] (cf. Supplementary Note 2 for a discussion on the energetic likelihood for the divacancy defect formation).

## Discussion

The growth of CIS under Cu excess leads to the formation of domains having Cu concentration beyond stoichiometry within the bulk of the films. The domains may consist of $Cu_2Se$ interspersed phase and/or Cu supersaturation in the form of Cu interstitials. Either way, this condition is compatible with previously reported metastable phase equilibria[8] and affects the surface chemical reactivity of CIS towards air, etchants, PDTs and solar cell finishing processes.

Exposure of CIS to air at room temperature leads to the spontaneous formation of $In_2O_3$ and $Cu_2Se$. This reaction occurs at a higher rate on Cu-rich than on Cu-poor CIS, presumably catalysed by the high mobility of available $Cu_i$ or by interspersed $Cu_2Se$ domains acting as seeds for further growth. KCN etching is needed to remove $Cu_2Se$, but it also imparts preferential leaching of Cu and Se from the underlying CIS lattice. The end result is a Cu and Se depletion from the CIS crystal lattice, responsible for the appearance of a $200 \pm 20$ meV defect from ADM measurements. The density of this defect is higher the higher the initial Cu/In ratio during CIS growth. Indeed, although standard CdS CBD is sufficient to remove the defect from Cu-poor material, the defect persists on Cu-rich material, where it causes front interface recombination and a large $V_{OC}$ deficit.

Specific chemical processes performed at the surface of (Cu and Se)-depleted Cu-rich CIS can eliminate the $200 \pm 20$ meV defect, hence reducing interface recombination and $V_{OC}$ deficit. These processes are typically associated to a high content of group

**Table 2 Parameters of solar cell devices comprising CIS absorbers with various compositions subject to surface treatments.**

| Sample | Efficiency (%) | $V_{OC}$ (mV) | $J_{SC}$ (mA cm$^{-2}$) | FF (%) |
|---|---|---|---|---|
| PW[71] | 12.8 | 446 | 42.0 | 68 |
| PB | 12.7 | 460.2 | 44.5 | 61.9 |
| SW[72] | 12.1 | 436.8 | 40.7 | 67.9 |
| SB | 11.7 | 434.5 | 38.5 | 69.7 |
| RW[60] | – | – | – | – |
| RB[60] | 7.0 | 355 | 42.1 | 46.8 |
| RB + Se-only[60] | 9.2 | 382 | 40.7 | 59.2 |
| RB + Se KF PDT[70] | 9.5 | 398 | 41.1 | 58.0 |
| RB + Zn(O,S)[60] | 6.7 | 399 | 41.0 | 40.9 |

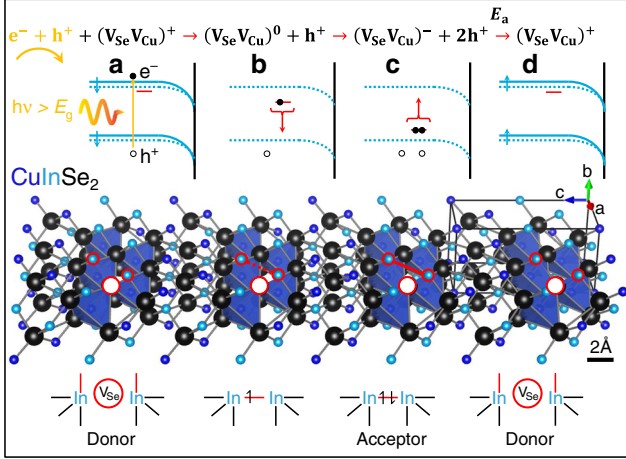

**Fig. 8 Amphoteric ($V_{Se}V_{Cu}$) divacancy as the culprit of the 200 meV defect.** Simplified model of light-induced metastability observed in Cu-rich CIS subject to cyanide etching (leading to preferential removal of Se and Cu well below the surface). **a** Before light perturbation the divacancy is in its donor configuration ($V_{Se}V_{Cu}$)$^+$ represented by the red state of the corresponding virtual hydrogen-like defect in the band diagram, and by the larger distance between the In atoms in the structural representation (bottom). **b** One photogenerated electron gets captured by the divacancy and partially occupies a bonding orbital between the two neighbouring In atoms, hence the divacancy assumes a neutral charge ($V_{Se}V_{Cu}$)$^0$. The bond being formed causes the In atoms to get closer, hence the energy level of the virtual hydrogen-like defect moves down. Electron trapping via chemical bond formation causes band flattening (dashed lines). TR-SPV reveals a lower degree of band flattening in Cu-rich CIS than in Cu-poor CIS, while PEC shows the slowly increasing photocurrent transient in Cu-rich CIS, instead of the decreasing photocurrent typical of Cu-poor CIS. **c** As the energy level gets closer to the valence band maximum, one electron is captured, leading to the full conversion of the divacancy into the acceptor state ($V_{Se}V_{Cu}$)$^-$ and to the largest structural displacement with respect to the donor configuration. **d** The complete In–In bond formation at point c entails a sizable activation energy for the cleavage of the acceptor via capture of two holes, hence the time constant associated to the surface charge reconstruction is on the order of tens of minutes at room temperature, as observed in TR-SPV.

VI elements of the periodic table (O and the chalcogens S and Se). Exposure of the absorbers to $O_2$ in the air at room temperature almost doubles the magnitude of electrochemical photocurrent recorded on semiconductor/electrolyte junctions with the distinct removal of a recombination pathway, as revealed by the shape of the photocurrent transient. Likewise, PDTs at higher temperature containing Se, and deposition of buffer layers with higher than standard concentration of S precursor reduce the shunt paths leading to improved $V_{OC}$, fill factor and efficiency of solar cells based on Cu-rich CIS absorbers.

The common denominator of such successful chemical processes is the group VI elements, which appear to exert a passivating effect on the Se-depleted CIS surface. It is deduced that the passivation is directed to Se vacancies in the near CIS surface, bringing the extinction of harmful Cu-Se divacancies.

This study reveals that the formation of vacancies in multinary chalcogenide materials can occur even at room temperature. The process is enabled by oxidation of the material near the phase boundary, as well as by chemical etchants. It may occur at the chalcogenide surface and extend tens of nanometres within the bulk of the material. Due to the relatively low bulk atomic diffusivities at room temperature, a metastable defected shell may form and persist over a timeframe of months or years (cf.

Supplementary Figs. 8 and 9), but can also be passivated in minutes with appropriate chemical surface treatments. The conversion of the chalcogenide into the oxide of the least noble cation upon air exposure is the thermodynamic driving force for preferential cation removal. Anion vacancy formation is then driven by preferential leaching of one of the cations by a complexant in solution, typically during surface etching. For chalcogenides at the edge of their existence region, vacancy formation may be exacerbated by a thermodynamically advantageous difference in electrochemical potential of the relevant species across the phase boundary.

## Methods

**Absorbers and device preparation.** Molybdenum coated soda lime glass was used as a substrate to fabricate polycrystalline CIS absorbers using physical vapour deposition in a molecular beam epitaxy (MBE) system. CIS was fabricated using a one-stage co-evaporation process at a substrate temperature of 530 °C, followed by unforced cooling to room temperature. The Cu/In ratio was varied through controlling the Cu and In fluxes. Absorbers grown under Cu deficiency were fabricated targeting a Cu/III ratio of <1 (CIS-P, 'Cu-Poor'); conversely a Cu/III ratio of more than one was targeted for those grown under Cu excess (CIS-R, 'Cu-Rich'). The elemental composition of the absorbers is determined by energy-dispersive X-ray spectroscopy with an acceleration voltage of 20 kV. This value represents the average composition of the absorber including Cu$_x$Se if present. After growth, the absorbers are subjected to an etching step to remove copper selenide secondary phases and residual oxides[37,38]. The absorbers were either immersed in KCN aqueous etching solutions or processed without etching (KCN-U, 'Unetched'). Two types of KCN conditions were employed: a 30 s etch in a 5 wt. % KCN (KCN-W, 'weak') or a 300 s etch in a 10 wt. % KCN (KCN-B, 'bold'). A CdS buffer layer was then deposited on top of the etched absorbers using chemical bath deposition before sputtering both a nominally un-doped and a biased zinc oxide window layer[63]. Lastly, nickel-aluminium contacting grids were evaporated by electron-beam deposition.

**Solar cell analysis.** The electrical parameters of the solar cells were extracted from the current–voltage (JV) measurements performed using a AAA Solar Simulator calibrated using a silicon reference solar cell, together with a JV source measure unit. JV analyses of CIS-RB devices were also measured as a function of annealing time at 60 °C and 80 °C. For this purpose, finished devices were annealed repeatedly on a hot plate for different durations, left to cool at room temperature and characterized in between each annealing period.

**Epitaxial growths and deliberate oxidation.** The growth of the 10 quintuple layer (QL) thick β-In$_2$Se$_3$ thin film was achieved by MBE on a c-sapphire substrate. The growth was performed at 550 °C under Se excess at rate of 1.5 QL min$^{-1}$. The thin film was left in air under dark conditions and its deliberate oxidation was studied by repeated Raman ex-situ analyses. The epitaxial CIS film was grown on a (100) GaAs wafer via metal-organic chemical vapour deposition by thermal decomposition at 470 °C of cyclopentadienyl-copper-triethyl phosphine, trimethyl-indium and ditertiarybutyl selenide. The relative gas partial pressures were adjusted to achieve a Cu-poor CIS composition (Cu:In equals 0.87, as assessed by EDS)[64]. The epitaxial CIS film was subject to graded oxidation at 550 °C by placing the 2 cm-long specimen inside a tube furnace along a gradient of incoming oxygen partial pressure ensured by adjusting the aperture of the end valve such that the pressure attained by the rotary pump of the apparatus did not exceed 100 mbar.

**Scanning electron microscopy.** The morphology of the absorbers was assessed with a Hitachi SU-70 field-emission SEM. Top-view SEM images were taken before and after KCN etching, as well as after the post-deposition treatment and HCl etchings, with a voltage of 7 and 20 kV with the intent to acquire compositional information at different depths. Cross-sectional SEM images were taken after removal of the excess Cu$_2$Se phase, at a voltage of 7 kV.

**Atom probe tomography.** The cryo-FIB preparation was carried out using a Gatan C1001 LN$_2$ cold module and an in-house designed adapter holder for the APT specimen. APT experiments were performed with a local electrode atom probe (LEAP 3000X HR, Cameca Instruments), applying laser pulses of 532 nm wavelength, 12 ps pulse length and an energy of 0.1 nJ per pulse at a repetition rate of 100 kHz. The specimen base temperature was about 60 K. Atom probe specimens were prepared utilizing a dual-FIB (FEI Helios Nanolab 600i) using a lift-out procedure[65].

**Raman spectroscopy.** The Raman spectra of the β-In$_2$Se$_3$ epitaxial layers were acquired with an excitation wavelength of 532 nm on a confocal microscope at ×50 magnification. Given that in-situ oxidation of the β-In$_2$Se$_3$ can be largely accelerated by the laser beam, the laser power was limited to 1 mW and an average of 10 acquisitions of 4 s duration was found appropriate to maximize signal to noise

ratio, while minimizing laser-induced oxidation. However, repeated exposure of the same spot over 5 min revealed clearly the decrease of signal. The intensity was normalized based on background signal.

**X-ray photoemission spectroscopy.** Surface chemical analysis of the absorbers was performed by XPS measurements using an ESCALAB 150Xi (Thermo Fisher Scientific) spectrometer and a monochromatic Al Kα radiation (1486.6 eV). Photoelectrons were detected parallel to the surface normal with a pass energy of 20 eV for the narrow scans. The XPS spectra were analysed using Avantage data processing software. Atomic concentration calculations and peak fitting were carried out after removing a Shirley-type background and by using the relative sensitivity factors provided by the the Avantage library. The sputtering of the samples was performed with Ar$^+$ ions ($E = 4$ keV) and with an angle of 45°.

**Secondary ion mass spectrometry.** SIMS depth profiling was performed with a Cameca SC-Ultra instrument using 1 keV Cs$^+$ ion bombardment. The analytical conditions used (primary ion beam, raster size) yield a sputtering rate of 0.20 nm s$^{-1}$. Secondary ion intensities were collected from an area of 60 μm in diameter. The SIMS profile data were normalized against the primary Cs$^+$ flux to account for beam fluctuations[64].

**Photoelectrochemical analysis.** PEC transients of the CIS films were recorded by holding the absorbers at −0.4 V vs. Ag/AgCl reference in a 0.2 M aqueous Eu$^{3+}$ solution, employing a three-electrode setup described previously[66]. Illumination was supplied by a 530 nm light-emitting diode (Thorlabs), subject to an asymmetric square-wave duty cycle. The PEC transients are described assuming three different kinetic constant of electron transfer: $k_{tr}$ (electron transfer from semiconductor to electrolyte) $k_{s.r.}$ (surface recombination) and $k_{b.t.}$ (bulk trap). The values of $k_{s.r.}$ are obtained from each individual photocurrent transient by fitting the data, according to Eq. (8) and assuming a $k_{tr} = 250$ s$^{-1}$ for all samples[47].

$$\frac{J(\infty)}{J(0)} = \frac{k_{tr}}{k_{tr} + k_{s.r.}} \quad (8)$$

**Intensity-modulated photocurrent spectroscopy.** IMPS analysis was performed on CIS samples of various compositions subject to different etching and subsequently coated with CdS. The acquisition conditions consist of negative polarization at −0.5 V vs. Ag/AgCl reference in a 0.2 equimolar aqueous solution of K$_4$[Fe (CN)$_6$] and K$_3$[Fe(CN)$_6$][47]. The CdS/ferro-ferricyanide junction yields durable and highly ideal photocurrent transients. This ensures the reproducible acquisition of IMPS spectra, unlike the CIS/Eu$^{3+}$ junction that is subject to irreversible changes of interface properties, as described within this study. The Nyquist plot is obtained from the Bode plot by means of the phase angle information ($\phi$), enabling the extraction of the real and imaginary photocurrents, as per Eq. (9).

$$J_{ph}(\omega) \cdot e^{-i\phi(\omega)} = J_{ph}(\omega) \cdot \{\cos[\phi(\omega)] - i\sin[\phi(\omega)]\} \quad (9)$$

**Time-resolved surface photovoltage.** KPFM was used to locally probe the photogenerated carriers by monitoring the CPD behaviour under illumination. KPFM experiments were performed in an ultra-high vacuum scanning probe microscope, Omicron Nanotechnology GmbH, controlled by Nanonis electronics and using a PtIr-coated cantilever ($f_0 = 167$ kHz). Amplitude modulation was used for the detection of the CPD with an ac bias of 400 mV at the second oscillation mode of the cantilever. Real-time KPFM measurements are not always suitable to characterize the carrier dynamics, since the KPFM controller time constant, of few tens of ms, is much slower than the time scale at which charge generation, separation and relaxation processes take place. Region I of the time domain response obtained on CIS-PB (in Fig. 6b) exemplifies this behaviour. Upon switching on and off a continuous-wave (cw) radiation source (635 nm laser, Thorlabs), the CPD shows a sharp increase followed by an abrupt decrease, but no information related to the time constants of the raise and decay can be extracted. Therefore, a time-dependent study of the SPV is performed by measuring an average SPV as a function of the frequency of a modulated light source[67]. The surface photovoltage is defined as: SPV = CPD$_{under illumination}$ − CPD$_{in dark}$. For the modulated light, a fast switched diode laser (PicoQuant FSL500) was externally triggered via user-defined signal patterns, avoiding any frequencies which could lead to artifacts due to frequency mixing[68]. The light sources have an optical power of ~4.5 mW (with an intensity of ~ 100 mW cm$^{-2}$) and an illumination angle of 28°, to ensure illumination of the sample under the tip and cantilever beam. The decay time ($\tau$) values are obtained from each individual frequency spectrum by fitting the data according to Eq. (10)[67].

$$V_{avg}(f) = V_{dark} + SPV \cdot D + \tau \cdot SPV \cdot \left(1 - e^{\frac{-(1-D)}{\tau \cdot f}}\right) \quad (10)$$

where $V_{dark}$ is the in dark potential, SPV is the surface photovoltage measured under cw illumination, $\tau$ is the SPV decay time, $f$ is the modulation frequency, and $D$ is the illumination duty cycle.

**Photoluminescence.** A home-built calibrated setup is used for the determination of PL properties. A continuous monochromatic illumination (660 nm) from a solid-state laser is used for excitation. The emitted PL is collected by an off-axis mirror, guided into a grating monochromator by an optical fibre and detected by an InGaAs-detector array.

**Admittance spectroscopy.** The temperature- and frequency-dependent ADM measurements were performed in the dark, after keeping the sample mounted in the dark at room temperature for one night, with a Precision LCR meter using closed-cycle helium cryostat in a temperature range of 320–50 K. ADM measurements are performed to study the electronic defect states in the semiconductor junction. The finite capture/emission time constants of such a defect level result in distinct steps in the ADM spectrum. The inflection frequency ($f_t$) of a capacitance step at a given temperature is determined by the defect response time and the thermal activation energy of the defect can be obtained from the slope of an Arrhenius plot of the temperature-dependent inflection frequencies (presented in Supplementary Fig. 7c). It is noteworthy that $f_t$ is commonly scaled with a factor $T^{-2}$ to account for implicit temperature dependences of the thermal velocity and effective density of states[48,52].

**Thermochemical computation.** The thermochemical treatment in Fig. 3a was elaborated using tabulated data[69]. The chemical potential of CuIIISe$_2$ is determined assuming the reported enthalpy of formation from the elements at 298 K[22,32,44] and an entropic contribution of ideal mixing ($\Delta S_{mix}$ equals $Rln(0.5)$), as per Eq. (11), and is consistent with previous studies[33].

$$G_{CuIIISe_2}(T) = \frac{G_{III_2Se_3}(T) + G_{Cu_2Se}(T)}{2}$$
$$- \left(H_{CuIIISe_2}(298) - \frac{H_{III_2Se_3}(298) + H_{Cu_2Se}(298)}{2}\right) - T\Delta S_{mix} \quad (11)$$

## Data availability

The datasets generated and/or analysed in the present study are available from the corresponding author on reasonable request.

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

## Acknowledgements

This contribution has been enabled by the Luxembourgish Fonds National de la Recherche (FNR) in the framework of the GALDOCHS project (Gas-phase alkali doping of chalcogenide semiconductors, C14/MS/8302176) and CURI-K project. INL and the European Commission are acknowledged for funding the Nano Train for Growth II project number 713640 through the Marie Curie Cofund programme. S. Sadewasser, N.N. and M.W. acknowledge funding from the European Union's Horizon 2020 research and innovation programme under grant agreement number 641004 (Sharc25). We acknowledge the Luxembourg Institute of Science and Technology (LIST) for SEM use and Mr. Brahime El Adib (LIST) for his skilful technical assistance in the SIMS analysis. Uwe Tezins, Andreas Sturm, and Christian Bross are acknowledged for their constant support in running the atom probe facilities at MPIE. T.S. acknowledges the support by the German Research Foundation (DFG) (Contract Number GA 2450/1-1). Lifeng Liu and Pedro Alpuim are acknowledged for giving access to the IMPS and Raman instruments. Professor Alberto Credi (Alma Mater Studiorum University of Bologna) is gratefully acknowledged for his insightful suggestions as chair of the 47th Conference of Inorganic Chemistry of the Italian Chemical Society held in Bari in September 2019, where part of this work was presented.

## Author contributions

D.C., H.E. and S. Siebentritt designed the research and experiments. H.E. and F.B. fabricated the absorbers and performed the post-deposition treatments. H.E. performed the JV and ADM characterizations. D.C. performed the defect chemical analysis, the PEC and IMPS characterizations. M.C. deposited the ten-layer $In_2Se_3$ stack by MBE and assisted during the Raman analyses. C.S. deposited the epitaxial CIS by MOVPE. N.N. and D.S. performed the TR-SPV analysis. T.S. performed cryo-FIB and APT analysis, and processed the 3D data. Other APT analyses were performed and processed by A.K. and O.C.-M. O.B. performed the XPS analysis. N.V. was in charge of the SIMS processing data. M.M. performed the baseline process of the cells and performed SEM and EDX characterization. M.S. performed solar cell characterization on devices subject to low-temperature annealing for different durations. M.W., F.B. and C.S. performed the PL characterizations. D.C., H.E. and S. Siebentritt wrote the paper. All authors (including S. Sadewasser, P.J.D. and D.R.) contributed to interpretation of measurement results and discussions for the manuscript.

## Competing interests

The authors declare no competing interests.
