## [Peer Review File · Nature Communications]

Reviewers' comments:

Reviewer #1 (Remarks to the Author):

The manuscript „chemical instability of chalcogenide surfaces near phase boundaries“ reports on the effects of oxidation on CuInSe₂ thin films grown with different Cu/In ratios. It is shown that different reactions take place at the surface depending on whether the film composition is Cu-poor or Cu-rich. Experimental results from Atom-probe, EDX, photoluminescence, PEC, SPV and Admittance. The manuscript describes phenomena which to a large extent have been known in the chalcopyrite community (as also referenced by the authors) while presenting some more detailed measurements with varying composition. The presentation of the figures is quite nice, with exception of some very complicated figures, such as Figure 4b, and the text is quite compact with so many acronyms that it is almost impossible for the reader to keep track which sample and method is being discussed in a given section of the text. The manuscript starts very general (as in the title) and then becomes very specific, focusing on details of CuInSe₂, thereby losing a general view on the subject aimed at in the title. Some of the measurements used to support the hypothesis, such as the PEC and SPV are not easy to interpret and leave the reader somewhat confused. I.e. the time scales in these measurements show transients in the microsecond to second range, for the former case then discussed as related to recombination, which is clearly way off the real carrier recombination times in this material. The solar cell results shown are not very coherent, and seem somewhat out of place. It is also not clear what the message arises from the determination of different activation energies from admittance. The authors discuss this as the creation and passivation of defects after oxidation, etching and exposure to sulfur, it does not really become clear what this has to do with surfaces near phase boundaries, as implied in the title of the manuscript. At different places incomplete data sets are discussed, e.g. XPS results for the surfaces of different samples are presented after oxidation, but no comparison is given to the pristine samples. The authors imply that vacancies are formed during the oxidation, but the methods used in the manuscript are not able to directly prove this. Overall the description is non-quantitative, i.e. the results are not modeled. With this regard, e.g. if the drop in Se measured by EDX in figure 4a is solely due to Se vacancies, the concentration of those defects would be huge. The authors discuss the changes in the bulk through surface reactions as extending 10s of nanometers into the bulk, however, the Se gradient measured in Fig 4a extends more than 1 micron into the bulk. Out of the reasons given above I do not see that this manuscript has the novelty or broad impact to warrant publication in nature communications, but rather recommend submission to a more specialized journal. Expanding the manuscript to make the presentation of results and discussion understandable to the reader (expert or non-expert) would be very helpful, e.g. I highly recommend leaving out most of the acronyms and reorganize the discussion to make it more coherent.

Reviewer #2 (Remarks to the Author):

This is a very detailed and comprehensive study of defect and secondary phase formation under exposure to air or cyanide etching of CuInSe₂ solar cell absorber films to study its chemical stability. A multitude of methods is employed to investigate the resulting film properties, covering the whole range from chemical composition, photoelectrochemical and optoelectronic properties up to the performance of complete solar cell devices. This study gives valuable new insights into fundamental material properties of CuInSe₂, which most likely can contribute to further improve the record efficiencies of solar cells based on this type of material. Additionally, this study also gives a nice example for how to approach similar studies for other compound semiconductor materials for high-efficiency solar cells. Therefore I consider it as suitable for publication in Nature Communications.

Before publication, however, I recommend consideration of the following points:

Cu-rich phase:

- * You refer to the metastable Cu-supersaturated phase from Gödecke and conclude that it may be possible to obtain this phase by growing under large Cu excess. This logic is not totally clear to me. Why would a "large" Cu excess help. Gödecke started at 850°C from a single phase with Cu-rich composition and reached the metastable phase by quenching. At your film growth temperature of 530 °C, however, in equilibrium you will have all the Cu-excess segregated as Cu₂Se according to Gödecke, independent from the amount of Cu excess. Also since you refer to the quenching procedure of Gödecke, I am missing information about the cooling rate after film deposition. Did you also do quenching? If not, why would you expect formation of the Cu-rich α_m phase that Gödecke obtained by quenching?
- * Further down, from "enhanced atomic diffusivity" you conclude on large concentrations of Cu interstitials in CIS-S. However, it is not clear to me how you know there is an enhanced diffusivity. If it is only an assumption then say so and give explain your assumption.
- * The platelets you see in APT look like Cu accumulation along lattice planes and/or grain boundaries, similar to the studies by Simsek Sanli. As far as I can see, increased Cu concentration is correlated with decreased In concentration, which qualitatively seems to be in accordance with the pseudo-binary phase diagram (and also in agreement with the results of Simsek Sanli). Hence, to me reaction (05) should then rather start from Cu (Cu_i)_z In(1-z) Se(2-z) instead of Cu (Cu_i) In Se₂. Can you explain how you conclude from the APT data that Cu in the platelets is chemically reduced and why you assume deviation from the Cu₂Se-In₂Se₃ line?
- * On page 4 you call all the films CIS-P, CIS-S, and CIS-R single-phased. But in the APT results CIS-R does not look single phased. How do you conclude that it is single-phases? It is stated that CIS-R is a Cu supersaturated CIS solution. According to APT in Fig. SI1, most of the CIS has a Cu/In < 1. The lines where Cu/In > 1 could be along planar defects, dislocations or grain boundaries, which I would not consider as a "phase". But if you do consider it as a phase, then would not be single-phased.

Oxidation:

- * Did you check if H₂O plays a role during the treatment in air, e.g. by using artificial dry air?
- * You state that the shoulder in PL points to the formation of V_{In}. It is not clear on what basis you draw this conclusion. Is this the only possible cause for the PL shoulder? I see that In from CIS is consumed for In-O formation and that hence formation of V_{In} in CIS seems likely, but how can you be sure? And how can you be sure that the PL shoulder is from V_{In}?
- * Eq. (04): The composition of your initial CIS phase does not necessarily lie on the Cu₂Se-In₂Se₃ line (only for 2y=z). Is this intentionally?
- * In eq. (05), do you really need O₂ to obtain Cu₂Se formation? Why not 2Cu(Cu_i)_z In Se₂ -> zCu₂Se + 2CuInSe(2-z/2) ? Did you try a similar treatment in vacuum instead of air? If you would not need O₂ this would mean that this reaction could take place spontaneously in the finished solar cell (which would be bad news).
- * Can you exclude that the 200 meV activation energy stems from Cu interstitials or Cu_{In} antisites? DFT calculations of Simsek Sanli et al. suggest that Cu accumulation at dislocations may cause deep defect states (DOI:10.1103/PhysRevB.95.195209).

Discussion:

At the end of the manuscript a broader discussion of the consequences of your findings for CIS or CIGS solar cell fabrication and the advantages and disadvantages of Cu-rich grown CIS vs Cu-poor CIS would be very valuable.

As a general remark, the manuscript is very dense, which is nice but makes it partially hard to read. At some points in the text I think the readability could be improved by adding some more explanations

or to refer to the Method section for further details (see e.g. some of the remarks below). At some points you write statements as if they were clear, where I have the impression that it is rather a speculation or assumption (see some of the remarks above and below). Please try to distinguish between these cases more clearly.

Some of the text in the Figures is so small that I could hardly read it on my printout. I recommend to increase the size of the figures. Headings and subheadings would help to navigate through the manuscript.

Additional minor remarks:

* In the introduction you state that "significant property changes can be observed with slight modifications of the synthesis...". Can you provide references for this statement?

* Introduction: "Different interface reactivity ..." -> I suggest to be more specific here. What kind of interfaces are you talking about?

* Introduction: You mention perovskite/CIGS tandems. It would be nice to cite work where this actually has been realized. See e.g. DOI: 10.1021/acseenergylett.9b00135, and references therein

* page 2: When you start to talk about "the films" (bottom paragraph of left column), it would be very helpful to give some details on the synthesis in the text (like 1-step deposition, growth temperature) and refer to the Method section for more details. Since you refer to the quenching process by Gödecke, it would be important to provide details on the cooling procedure.

* page 2: What exactly do you mean by "nominally stoichiometric material". Please explain.

* You do not refer to Fig. SI1 and SI3 in the text. In Fig. SI1 a unit for the x-axis is missing and the data range partly cuts off the In signal. This should be corrected.

* page 3, top: What exactly do you mean by "purely Cu-rich CIS"? Cu-rich grown and then etched? Or does this refer to the fact that you do not start with Cu-poor CIS? Please be more specific here.

* Fig. 2a: It would improve readability if you note the corresponding equation numbers (01) to (03) in the graph.

* page 4: Please give experimental details of the fabrication of the Cu-poor epitaxial CIS film and the graded oxidation procedure.

* page 4: What do you mean by "Under thermodynamic conditions"? Do you mean "thermodynamic equilibrium"?

* page 5: You write "CIS-SB shows a PEC transient that is a convolution between those of CIS-PB and CIS-RB." I cannot see how CIS-SB is a convolution of CIS-PB and CIS-RB. Is this the right wording?

* Fig. 3a: The terms PB, ... should be introduced in the figure caption.

* Fig. 3b: Is the data shown for region I and II from two different measurements? I: real-time, II: modulated illumination. You refer to the "spectroscopic curve" in Fig. 3b, but I only see data vs. time.

* page 6: You write "KCN-W is unable to remove the entire Cu-Se phases" -> Do you have a proof for this statement? If not, mention that this is an assumption.

* For the SIMS you measured PW and RB. Why do you compare different composition AND different etching? This way it is not clear whether the difference stems from variation of the deposition process, or from the etching process. You also mention depth profiles of CIS-RB. Where is it shown? It would help to show it in the SI and refer to it.

* page 6: "The decrease of Cu content in CIS-RB does not imply the formation of a sub-stoichiometric Cu surface, as shown by the Cu/In SIMS ratio (inset)." -> This is not clear to me. I see that the Cu/In ratio decreases at the surface to the level of the bulk of CIS-PW. Since the bulk of PW should be Cu-poor, I would conclude that the surface of RB is also Cu-poor.

* page 7: "... the formation of vacancies in multinary chalcogenide materials can be a spontaneous process..." -> You see a reaction when you offer O₂ as a reactant. Can this really be called "spontaneous"?

Reviewer #3 (Remarks to the Author):

The authors studied the relationship between the surface chemistry of CuInSe₂ and its device characteristics using photoluminescence spectra, impedance analysis, photo-electrochemical responses under its different surface states. The conclusion that Se deficiency in CuInSe₂ is a problem on Voc deficit is reasonable. I partially understood the author's explanations while it needs to overcome some of the issues to be addressed before the publication.

(1) The authors concluded that photoluminescence from the deep level (0.9 eV) shown in Fig.2(c) originates from indium vacancy, V_{In}, and explained the reason using the chemical reaction formula (04). I think this interpretation is incorrect. According to the theoretical calculation of the defect formation energy using cluster structure (<https://doi.org/10.1016/j.jpics.2005.09.067>), the formation of V_{In} is hard to occur. Actually, In-Se chemical bonds form the main backbone of the chalcopyrite crystal structure while Cu is easy to migrate. Thus, nobody directly observed indium vacancy. Nevertheless, the authors insist high density of V_{In} introduces 0.9 eV luminescence peak. (I cannot trust it.) If the authors want to claim that indium vacancy originates luminescence peak, then some experimental evidence is required, such as detection it by using positron annihilation.

I guess it is possibly caused by In on Cu antisite, In_{Cu}, related defect.

(2) Why did the author not show the result of stoichiometric CuInSe₂, CIS-S, in Fig 3 and 4 (for instance with the depth profile of Se suggested in Fig. 4(a))? It is well known that the properties of Cu-rich CuInSe₂, CIS-R, are different from those of Cu-poor CuInSe₂, CIS-P, due to its phase diagram. CIS-R must include many Cu₂Se components in the films initially (before KCN etching). I guess the property of CIS-S is somewhat resemble that of CIS-P. In contrast, Cu vacancy, V_{Cu}, concentration affects migration of Na from soda lime glass. (Indeed, the point of Na concentration was not discussed in the manuscript. This is strange.)

Thus the results on CIS-S must be helpful to understand the underlying physics in detail.

(3) I agree with the author's suggested model (in Fig.4(b)) that (i) V_{Se} is firstly passivated by air (O_2 or H_2O ?) and (ii) form $Cu_2Se+In_2O_3$ by introducing excess oxygen. But I want to ask the authors why V_{Se} did mainly cause V_{oc} deficit. Actually, V_{Se} is located at 80 meV from conduction band minimum (shallow donor). It may act as self-compensation center. But since the level is too shallow, I think its impact on V_{oc} is not so big. Moreover, V_{Se} is easy to combine with V_{Cu} to form di-vacancy (<https://doi.org/10.1016/j.tsf.2016.02.057>), and many papers suggested the possibility of formation of di-vacancy, which energy level is located deeper in the gap. Why did the authors exclude the discussion on di-vacancy?

(4) Why did not the author mention the efficiency of the KCN etching on CIS-R to remove Cu_2Se ? Referring to Fig. 1(b), it seems the CIS-R seems to have been grown out of the CIS metastable region and in the $Cu_2Se + CIS$ region. It is moreover shown later in the article that a weak KCN etching can leave a significant amount of Cu_2Se on the sample, which has a strong effect on its properties. I think it would be interesting for the discussion to include the amount of Cu_2Se (if any) remaining after the KCN etching on samples presented at the beginning of the article.

COMMENTS TO AUTHOR:

Reviewer #1:

(1) *The manuscript „chemical instability of chalcogenide surfaces near phase boundaries“ reports on the effects of oxidation on CuInSe₂ thin films grown with different Cu/In ratios. It is shown that different reactions take place at the surface depending on whether the film composition is Cu-poor or Cu-rich. Experimental results from Atom-probe, EDX, photoluminescence, PEC, SPV and Admittance. The manuscript describes phenomena which to a large extent have been known in the chalcopyrite community (as also referenced by the authors) while presenting some more detailed measurements with varying composition. The presentation of the figures is quite nice, with exception of some very complicated figures, such as Figure 4b, and the text is quite compact with so many acronyms that it is almost impossible for the reader to keep track which sample and method is being discussed in a given section of the text.*

We agree that the manuscript is rather compact, containing many acronyms. The acronyms themselves are chosen intuitively and should be easy to recall, once the reader refers to the legend in the experimental section. However, the original text was indeed too condensed. **We took this opportunity to expand the text – especially in those parts where different samples are discussed – taking care to highlight the rationale that connects together the different experiments.**

(2) *The manuscript starts very general (as in the title) and then becomes very specific, focusing on details of CuInSe₂, thereby losing a general view on the subject aimed at in the title. [...see point (3)...], it does not really become clear what this has to do with surfaces near phase boundaries, as implied in the title of the manuscript.*

We realised that the original title may sound misleading. The concept of *phase boundary* refers to phase equilibria, and specifically to the regions of a phase diagram whose chemical composition leads to incipient segregation of a secondary phase. In the intended interpretation for the title, it is the chalcogenide (and not the surfaces) that is located near phase boundaries. In other words, we do not refer to the *physical* boundaries of the phase itself, such as *grain* boundaries.

An appropriate alternative wording to “*phase boundaries*” would be “*domain of single-phase existence*”. However, this terminology would narrow down the potential readership. Therefore, to avoid any confusion, **we opt for the following title: “Chemical instability of chalcogenide surface: technological impact well beyond the surface”**. The new title also reveals explicitly the novelty of the findings, i.e. that the metastable defect population (formed as a result of oxidation and post-processing, even at room temperature) occurs way deeper than the surface, unlike generally observed in other systems such as III-V semiconductors.

(3) *Some of the measurements used to support the hypothesis, such as the PEC and SPV are not easy to interpret and leave the reader somewhat confused. I.e. the time scales in these measurements show transients in the microsecond to second range, for the former case then discussed as related to recombination, which is clearly way off the real carrier recombination times in this material.*

The reviewer probably refers to the time constants associated with the PEC and SPV transients observed with Cu-rich CIS subject to bold KCN etching in Fig. 3a-b.

We realize that the text does not give justice to these findings, which we find particularly interesting. Thus, the novelty may have been difficult to grasp.

Firstly, it is important to point out that the physical phenomena happening during the illumination and relaxation at semiconductor/electrolyte interfaces are very different from those happening at semiconductor/vacuum interfaces. In the former case, electron transfer occurs from the semiconductor to redox species dissolved in a liquid medium, while in the latter case no electrons escape from the semiconductor.

The PEC electron transfer kinetics depends on the extent of energetic overlap between band edge and distribution of redox states in solution, and on the mechanism of electron transfer to the oxidised species in solution (i.e. outer-sphere or inner-sphere [1]). Typical rate constants are on the order of

10^2 s^{-1} . These and recombination phenomena happening at the semiconductor's surface are exclusive to PEC, so no comparison to SPV can be made (nor it is claimed) in such two cases.

However, the PEC findings are exceptional because they also reveal for the first time a recombination phenomenon happening in the bulk, which results in complex transients never reported before. Normally, charge carrier phenomena happening in the bulk of semiconductors are not resolved by PEC because their time constant is far too short compared to the acquisition rate capabilities of standard potentiostats. However, in this specific case, the time constant associated to traps present in the bulk (or near the surface) of Cu-rich CIS subject to bold KCN etching happens to be so exceptionally long that it yields a sizable time domain contribution in the PEC transient. Here, by relating this effect to the SPV transients, we were able to identify its common solid-state nature.

Quantifying the time constant associated to the bulk recombination from the PEC transients is impossible because of the limited number of data points that can be acquired on a time domain scale with the potentiostat employed. Nevertheless, in order to unambiguously confirm that this recombination route is the same as the one observed from the SPV transients, we have decided to perform intensity-modulated photocurrent spectroscopy (IMPS) measurements, which are more suited for the detection and quantification of time constants associated with recombination phenomena, via analysis of the frequency domain [2]. A time constant of ca. 2 microseconds is identified by IMPS for both Cu-poor and Cu-rich films subject to bold KCN etching, **which is in line with the value determined from SPV for Cu-rich films subject to bold KCN etching**. The extent of recombination in RB is ca. one order of magnitude higher than in PB, in line with the PEC results. Furthermore, these experimental findings corroborate a theoretical model published by Lany and Zunger in 2006 [3]. The results are included in the supplementary information and the text has been expanded accordingly, including with a new figure containing an adapted simplification of said model placed at the end of the manuscript. We hope that the reviewer's concern is now addressed.

(a) Bode plot of photocurrent and phase as a function of illumination frequency. (b) Corresponding Nyquist plot to enable the extraction of the time constants for recombination, according to Eq. 08.

(a) Schematic energy level diagram for the $(V_{Se}-V_{Cu})$ vacancy complex in CIS. (b) Configuration coordinate diagrams showing the energies of the different charge states of $(V_{Se}-V_{Cu})$, relative to $(V_{Se}-V_{Cu})^+$, which is the initial configuration of the optical cycle for the red illumination effect. The total charge is conserved by addition of free electrons. The green and red squares indicate the relaxed In-In distance of the donor and acceptor configurations of $(V_{Se}-V_{Cu})$, respectively. (a) and (b) are adapted

from [S. Lany and A. Zunger, "Light- and bias-induced metastabilities in Cu(In,Ga)Se₂ based solar cells caused by the (V_{Se}-V_{Cu}) vacancy complex," *Journal of Applied Physics*, vol. 100, no. 11, p. 113725, Dec. 2006], with the kind permission of AIP Publishing and Stephan Lany. (c) New Fig. 8 in revised manuscript – *Amphoteric (V_{Se}V_{Cu}) divacancy as the culprit of the 200 meV defect – Simplified model of light-induced metastability observed in Cu-rich CIS subject to cyanide etching (leading to preferential removal of Se and Cu well below the surface)*. (a) Before light perturbation the divacancy is in its donor configuration (V_{Se}V_{Cu})⁺ represented by the red state of the corresponding hydrogen-like defect in the band diagram, and by the two partially unbound In atoms in the chemical representation. (b) One photogenerated electron gets captured by the divacancy and partially occupies a bonding orbital between the two neighbouring In atoms, hence the divacancy assumes a neutral charge (V_{Se}V_{Cu})⁰. The bond being formed causes the In atoms to get closer, hence the energy level of the hydrogen-like defect moves down. Electron trapping via chemical bond formation causes (I) the lower degree of band flattening (dashed lines) compared to the Cu-poor case, as observed by TR-SPV, and (II) the slowly increasing photocurrent transient, instead of the decreasing photocurrent typical of Cu-poor CIS, as observed by PEC. (c) As the energy level gets closer to the valence band maximum, one electron is captured, leading to the full conversion of the divacancy into the acceptor state (V_{Se}V_{Cu})⁻ and to the largest structural displacement with respect to the donor configuration. (d) The complete In-In bond formation at point (c) entails a sizable activation energy for the cleavage of the acceptor via capture of two holes, hence the time constant associated to the surface charge reconstruction is on the order of tens of minutes at room temperature, as observed in TR-SPV.

(4) *The solar cell results shown are not very coherent, and seem somewhat out of place. It is also not clear what the message arises from the determination of different activation energies from admittance. The authors discuss this as the creation and passivation of defects after oxidation, etching and exposure to sulfur, it does not really become clear what this has to do with surfaces near phase boundaries, as implied in the title of the manuscript.*

The reviewer rightly points out that this part looks somehow incomplete. **The effect** of surface defects induced by oxidation, etching and postdeposition treatments on solar cell characteristics and admittance measurements **are now discussed more thoroughly**. The answer at point (2) should clarify the concern on phase boundaries. A new schematic energy diagram is also included, giving reference to the relevant study of Lany and Zunger (*Journal of Applied Physics* 100 113725, 2006) [3]. Typical low-temperature photoluminescence spectra of Cu-poor and Cu-rich films are now also included right at the beginning to better contextualise the research puzzle between material properties and device performance.

(5) *At different places incomplete data sets are discussed, e.g. XPS results for the surfaces of different samples are presented after oxidation, but no comparison is given to the pristine samples.*

Unfortunately, we are not equipped to transfer samples freshly obtained by co-evaporation or after etching into the XPS chamber without contact to air. Since XPS is an extremely surface-sensitive technique, **it is virtually impossible for us to obtain the XPS signature of pristine samples**.

(6) *The authors imply that vacancies are formed during the oxidation, but the methods used in the manuscript are not able to directly prove this. Overall the description is non-quantitative, i.e. the results are not modeled. With this regard, e.g. if the drop in Se measured by EDX in figure 4a is solely due to Se vacancies, the concentration of those defects would be huge. The authors discuss the changes in the bulk through surface reactions as extending 10s of nanometers into the bulk, however, the Se gradient measured in Fig 4a extends more than 1 micron into the bulk.*

We agree with the reviewer that the methods employed alone do not provide a direct proof for the presence of Se vacancies at or near the surface of our films. We are not aware nor of a single technique that could unambiguously provide such a proof. Rather, this research question requires evidence from a large ensemble of techniques, but the different pieces of information make it nearly impossible to construct a credible *quantitative* model, that is why we limited our claims to a more *qualitative*

description (yet containing some quantified information). The results of the techniques deployed makes us relate our findings to the presence of Se vacancies. One of these techniques is the depth profiling supplied in Fig. 4a that the reviewer assigns to EDX, but is actually based on secondary ion mass spectrometry (SIMS). Unlike EDX analysis, SIMS does not provide quantitative determination of elemental compositions, unless suitable standards are available. Because of the very surface instability studied in this research, in reality suitable chalcopyrite standards **cannot** exist. Nevertheless, SIMS is very sensitive to compositional differences, making it very suited for qualitative comparisons of the two samples in Fig. 4a. By no means should the drop of Se signal be related quantitatively to a corresponding concentration of Se vacancies. **In order to make this point very clear, we have now enclosed additional EDX analysis of the two samples using low electron acceleration voltage, showing that the Se ratio is indeed almost the same, except a subtle (but meaningful) effect, in line with the hypothesis brought forward.**

(7) Out of the reasons given above I do not see that this manuscript has the novelty or broad impact to warrant publication in nature communications, but rather recommend submission to a more specialized journal. Expanding the manuscript to make the presentation of results and discussion understandable to the reader (expert or non-expert) would be very helpful, e.g. I highly recommend leaving out most of the acronyms and reorganize the discussion to make it more coherent.

The relevance of the manuscript's findings to another chalcogenide material has now been extended with additional experimental evidence on epitaxial In_2Se_3 , relating also to the thermochemical data already shown in the original manuscript This 2D chalcogenide has potential applications in future atomically-thin devices and heterostructures. Therefore, the revised manuscript has a broader appeal to communities beyond PV.

We have also followed the referee's suggestion to edit and expand the manuscript in order to improve clarity of presentation and have reorganized the discussion accordingly. The text has been divided in more manageable sections enabling a more logic flow.

Reviewer #2:

This is a very detailed and comprehensive study of defect and secondary phase formation under exposure to air or cyanide etching of CuInSe_2 solar cell absorber films to study its chemical stability. A multitude of methods is employed to investigate the resulting film properties, covering the whole range from chemical composition, photoelectrochemical and optoelectronic properties up to the performance of complete solar cell devices. This study gives valuable new insights into fundamental material properties of CuInSe_2 , which most likely can contribute to further improve the record efficiencies of solar cells based on this type of material. Additionally, this study also gives a nice example for how to approach similar studies for other compound semiconductor materials for high-efficiency solar cells. Therefore I consider it as suitable for publication in Nature Communications.

We thank the referee for the kind remarks and for the exceptional effort devoted to review our manuscript. We share his/her view that the findings should help future improvements of CIGS solar cells. Given the advanced status of the technology, it seems likely that each remaining overlooked area can lead to breakthrough improvements.

Before publication, however, I recommend consideration of the following points:

Cu-rich phase:

(1) * You refer to the metastable Cu-supersaturated phase from Gödecke and conclude that it may be possible to obtain this phase by growing under large Cu excess. This logic is not totally clear to me. Why would a "large" Cu excess help. Gödecke started at 850°C from a single phase with Cu-rich composition and reached the metastable phase by quenching. At your film growth temperature of 530 °C, however, in equilibrium you will have all the Cu-excess segregated as Cu_2Se according to Gödecke, independent from the amount of Cu excess. Also since you refer to the quenching procedure of

Gödecke, I am missing information about the cooling rate after film deposition. Did you also do quenching? If not, why would you expect formation of the Cu-rich alpha_m phase that Gödecke obtained by quenching?

We understand the concern raised by the referee. Indeed, the Cu-supersaturated phase reported by Gödecke was obtained through an unusual thermal history. Quenching (i.e. kinetics) certainly plays a role when it comes to forming metastable phases, while in our case the samples were left to cool naturally to room temperature (as now specified in the text). However, the reported existence of this metastable phase means that - from the thermodynamic point of view - there exists a region of relative stability (a relative minimum in the Gibbs free energy landscape) that *may be* reached via alternative sample histories. It cannot be excluded that the growth under Cu excess is one of such routes.

Importantly, the equilibrium diagram determined by Gödecke was obtained with samples subject to step cooling at artificially low rates, far slower than those encountered after film growth. Therefore, it seems likely that true thermodynamic conditions (or in any case, conditions close to those reported for the equilibrium phase diagram by Gödecke) do not necessarily occur in standard thin film processing. The hypothesis of this study is precisely that partial formation of this metastable phase may occur even under *seeming* equilibrium at 530 °C. As also stated in the manuscript, highly efficient CIGS is grown as a Cu-poor material with Cu/(In+Ga) ratios as low as 0.8 (21.6 at. % Cu) [4], [5], *i.e. potentially* in a CIGS Cu-poor metastable homogeneity region.

According to Yuan et al. [6] the solubility of Na in CIGS is higher at growth temperature than at room temperature. In much the same way, Cu solubility in CIGS at growth temperature is likely to exceed the solubility limit at room temperature, with Cu atoms most probably capable of being incorporated as interstitials, in analogy to the Na case [7], [8]. In our manuscript we do not claim that the extent of metastable phase in CIS films grown under Cu excess equals the solubility range reported by Gödecke et al.. Even solubilities as low as one tenth of that reported in the metastable diagram of Gödecke are sufficient to justify the formation of a Cu supersaturated CIS phase, one that behaves differently when exposed to air or chemical etchants or during sample preparation for atom probe tomography.

In order to address the referee's concern, the abovementioned considerations are now explicitly included in the text and an additional schematic illustration is drawn to explain the hypothesis.

(2) Further down, from "enhanced atomic diffusivity" you conclude on large concentrations of Cu interstitials in CIS-S. However, it is not clear to me how you know there is an enhanced diffusivity. If it is only an assumption then say so and give explain your assumption.*

This was indeed a silent implication based on the observation of enhanced grain growth in CIS-S despite the absence of Cu-Se phases. The text is now edited accordingly.

(3) The platelets you see in APT look like Cu accumulation along lattice planes and/or grain boundaries, similar to the studies by Simsek Sanli. As far as I can see, increased Cu concentration is correlated with decreased In concentration, which qualitatively seems to be in accordance with the pseudo-binary phase diagram (and also in agreement with the results of Simsek Sanli). Hence, to me reaction (05) should then rather start from Cu (Cu_i)z In(1-z) Se(2-z) instead of Cu (Cu_i) In Se2. Can you explain how you conclude from the APT data that Cu in the platelets is chemically reduced and why you assume deviation from the Cu2Se-In2Se3 line?*

New APT compositional depth profiles are included in Fig. 2b, showing that the bulk of CIS-R is actually off-stoichiometric, with both Cu and In above 25 at. % and Se below 50 at. %. Having said that, we have changed Eq. (5) to the following:

However, we refer the reviewer to point 7 below for additional considerations.

(4) On page 4 you call all the films CIS-P, CIS-S, and CIS-R single-phased. But in the APT results CIS-R does not look single phased. How do you conclude that it is single-phases? It is stated that CIS-R is a*

Cu supersaturated CIS solution. According to APT in Fig. S11, most of the CIS has a Cu/In < 1. The lines where Cu/In > 1 could be along planar defects, dislocations or grain boundaries, which I would not consider as a "phase". But if you do consider it as a phase, then would not be single-phased.

Thinking about this observation, we could also imagine that CIS-R is not a metastable phase, but the decomposition product of the metastable phase obtained at high temperature. Again, along the same line of the research by Yuan et al. on Na doping [6], the excess Cu incorporated during the growth could migrate to planar defects, dislocations and/or grain boundaries during cool down or at room temperature. It could then be the excess Cu in those regions that leads to the different chemical behaviour of CIS-R towards air exposure and etchants. This alternative hypothesis in line with the referee's comment is now included in the manuscript.

Oxidation:

(5)* *Did you check if H₂O plays a role during the treatment in air, e.g. by using artificial dry air?*

We have not checked that, but this is a good point. When assessing and reporting the behaviour of the epitaxial gallium selenide 2D material towards air, we have come across a recent report discussing the role played by water during oxidation. **The reference to this relevant study [9] is now included in the manuscript.**

(6)* *You state that the shoulder in PL points to the formation of V_{In}. It is not clear on what basis you draw this conclusion. Is this the only possible cause for the PL shoulder? I see that In from CIS is consumed for In-O formation and that hence formation of V_{In} in CIS seems likely, but how can you be sure? And how can you be sure that the PL shoulder is from V_{In}?*

We refer to the following study for a detailed answer to the referee's question. Babbe, F., Elanzeery, H., Wolter, M. H., Santhosh, K. & Siebentritt, S. The hunt for the third acceptor in CuInSe₂ and Cu(In,Ga)Se₂ absorber layers [10].

(7)* *Eq. (04): The composition of your initial CIS phase does not necessarily lie on the Cu₂Se-In₂Se₃ line (only for 2y=z). Is this intentionally?*

The phase diagram in Fig. 1 is very helpful to understand and interpret chemical equilibria. However, it is a construct limited to pseudobinary compositions containing two phases at most. Deviations from such compositions are possible in reality, especially as a result of chemical reactions involving foreign components leading to changes of oxidation states and/or formation of new phases. By keeping the initial CIS composition as indicated in Eq. (04), we accept the possibility for the concentration of copper vacancies and indium on copper antisites to vary according to the respective formation energies that depend on the chemical potentials of the components and on the Fermi level.

(8)* *In eq. (05), do you really need O₂ to obtain Cu₂Se formation? Why not 2Cu(Cu_i)_z In Se₂ -> zCu₂Se + 2CuInSe(2-z/2) ? Did you try a similar treatment in vacuum instead of air? If you would not need O₂ this would mean that this reaction could take place spontaneously in the finished solar cell (which would be bad news).*

We think that the formation of indium oxide provides the thermodynamic driving force for the reaction to happen, but it is the large pool of Cu interstitials that makes it occur at a higher rate compared to Cu-poor material, due to a lowering of the reaction energy barrier (i.e. acting kinetically). An energy diagram is now included to illustrate the proposed mechanism. The rationale for the choice of stoichiometry in reaction (05) stems from the assignment of indium vacancies as discussed at point 6. Having said that, we cannot exclude that Cu-rich CIS films stored for long time under oxygen-free environment incur slowly the reaction suggested by the referee. In an attempt to provide further insights into this question, we have measured the electrical characteristics of more than 3 years old Cu-poor and Cu-rich devices stored in the desiccator and compared them to the original values. The results do not show large differences between the two cases, as appended in the next page.

JV analyses of Cu-poor (a,b) and Cu-rich (c,d) CIS devices as measured on the day of cell fabrication (a,c) and after more than 3 years of storage in the desiccator (b,d).

We anticipate that the figure may raise additional comments by the referee, especially pertaining the slight performance improvement observed for the Cu-rich CIS device. Therefore, we have decided to perform a low-temperature annealing study in an attempt to simulate accelerated room temperature ageing of Cu-rich devices and test if the slight performance improvement could be reproduced. The figure below shows the average JV parameters of five devices annealed at 60 °C and 80 °C as a function of the total annealing time. This extra work justifies the inclusion of Mohit Sood as an additional author for this manuscript.

(9)* *Can you exclude that the 200 meV activation energy stems from Cu interstitials or Cu_{In} antisites? DFT calculations of Simsek Sanli et al. suggest that Cu accumulation at dislocations may cause deep defect states (DOI:10.1103/PhysRevB.95.195209).*

Thank you for pointing to this research. We cannot exclude that the 200 meV defect stems from Cu interstitials and/or Cu_{In} antisites, so we now refer to the local density of states calculated by DFT in Simsek Sanli et al. [11]. However, if these were the sole origins of the defect, it would remain unclear why KCN etching relates to the appearance of such a defect. Therefore, we think that the cited paper does not affect the conclusions of our manuscript, but we remain open to other interpretations the referee may have.

Discussion:

(10) *At the end of the manuscript a broader discussion of the consequences of your findings for CIS or CIGS solar cell fabrication and the advantages and disadvantages of Cu-rich grown CIS vs Cu-poor CIS would be very valuable.*

The conclusions of the manuscript have been largely revised in light of the literature that is now cited.

(11) *As a general remark, the manuscript is very dense, which is nice but makes it partially hard to read. At some points in the text I think the readability could be improved by adding some more explanations or to refer to the Method section for further details (see e.g. some of the remarks below). At some points you write statements as if they were clear, where I have the impression that it is rather a speculation or assumption (see some of the remarks above and below). Please try to distinguish between these cases more clearly.*

Some of the text in the Figures is so small that I could hardly read it on my printout. I recommend to increase the size of the figures. Headings and subheadings would help to navigate through the manuscript.

We agree on these remarks, which are also in line with points 1 and 7 from Referee#1. **The text and the figures have been expanded and modified throughout the manuscript to improve readability.**

Additional minor remarks:

(13)* *In the introduction you state that "significant property changes can be observed with slight modifications of the synthesis...". Can you provide references for this statement?*

Perhaps the most prominent case where slight modifications of the synthesis conditions such as composition, thermal history and mechanical stress bring about large variations of the physical properties is that of steels with the iron-carbon system. We added a reference in this regard [12].

(14)* *Introduction: "Different interface reactivity ..." -> I suggest to be more specific here. What kind of interfaces are you talking about?*

The intent here is to introduce the issue of potential reactions happening at surfaces, grain boundaries and back contacts. See for example [13]. The text has been amended accordingly.

(15)* *Introduction: You mention perovskite/CIGS tandems. It would be nice to cite work where this actually has been realized. See e.g. DOI: 10.1021/acenergylett.9b00135, and references therein*

The paper by Jost et al. is now cited in the introduction [14].

(16)* *page 2: When you start to talk about "the films" (bottom paragraph of left column), it would be very helpful to give some details on the synthesis in the text (like 1-step deposition, growth temperature) and refer to the Method section for more details. Since you refer to the quenching process by Gödecke, it would be important to provide details on the cooling procedure.*

The referee's point is now addressed. Thanks to this comment we also realised that the word "films" was missing when we first referred to the samples fabricated in our study.

(17)* *page 2: What exactly do you mean by "nominally stoichiometric material". Please explain.*

In this case we mean a CIS film grown under Cu excess and later subject to chemical removal of the Cu-Se excess, which is generally accepted to leave only CIS material with composition at the edge of the homogeneity region at room temperature. This is now specified in the text to avoid any confusion with the film actually grown under nearly stoichiometric conditions.

Only later in the manuscript is the secondary effect of such etching process discussed with an eye on the near-surface defect distribution.

(18)* *You do not refer to Fig. S11 and S13 in the text. In Fig. S11 a unit for the x-axis is missing and the data range partly cuts off the In signal. This should be corrected.*

The supplementary figures (now referred to in the manuscript) have been largely revised, also based on new APT analyses, justifying the inclusion of Torsten Schwarz as an additional author.

(19)* *page 3, top: What exactly do you mean by "purely Cu-rich CIS"? Cu-rich grown and then etched? Or does this refer to the fact that you do not start with Cu-poor CIS? Please be more specific here.*

Clarified. We mean CIS grown under Cu-rich conditions from which the excess Cu-Se phase has been removed. We clarified accordingly.

(20)* *Fig. 2a: It would improve readability if you note the corresponding equation numbers (01) to (03) in the graph.*

(21)* *page 4: Please give experimental details of the fabrication of the Cu-poor epitaxial CIS film and the graded oxidation procedure.*

Done.

(22)* *page 4: What do you mean by "Under thermodynamic conditions"? Do you mean "thermodynamic equilibrium"?*

Yes, we replaced the word "conditions" with "equilibrium". The reason why we opted for "conditions" is that generally speaking, processes may occur under thermodynamic equilibrium or may instead be dominated by kinetic effects, in which case it is often stated that the system is under "kinetic control" or follows "kinetic conditions".

(23)* *page 5: You write "CIS-SB shows a PEC transient that is a convolution between those of CIS-PB and CIS-RB." I cannot see how CIS-SB is a convolution of CIS-PB and CIS-RB. Is this the right wording?*

By convolution we meant that **the transient of CIS-SB can be described by a linear equation involving the transients of CIS-PB and CIS-RB**. A careful look at the CIS-SB curve from the time when the illumination is turned off onwards shows that the current becomes more positive, reaches a maximum and then becomes more negative. In the case of CIS-PB the current only becomes more positive with time, while in the case of CIS-RB the current only becomes more negative with time. **Hence a more appropriate wording is chosen to describe the behaviour of CIS-SB: linear combination.**

(24)* *Fig. 3a: The terms PB, ... should be introduced in the figure caption.*

Specified.

(25)* *Fig. 3b: Is the data shown for region I and II from two different measurements? I: real-time, II: modulated illumination. You refer to the "spectroscopic curve" in Fig. 3b, but I only see data vs. time.*

No, the data in region I and II comes from the very same measurement, and it simply relates to the CPD recorded as a function of time, during the illumination modulation exemplified by the yellow-highlighted area. **We acknowledge that the wording may have created some confusion. As we meant "time domain response" rather than "spectroscopic curve" we have changed the text accordingly.**

(26)* *page 6: You write "KCN-W is unable to remove the entire Cu-Se phases" -> Do you have a proof for this statement? If not, mention that this is an assumption.*

We supply the corresponding evidence acquired by SEM/EDS as part of a new figure in the manuscript (new Fig. 5).

(27)* *For the SIMS you measured PW and RB. Why do you compare different composition AND different etching? This way it is not clear whether the difference stems from variation of the deposition process, or from the etching process. You also mention depth profiles of CIS-RB. Where is it shown? It would help to show it in the SI and refer to it.*

We agree that the data currently available does not enable to separate obviously the effects of etching from those due to the different growth. Since SIMS analyses are very expensive, we limited ourselves to the two most explicative samples of this study (CIS-PW and CIS-RB). **For completeness we have now performed SIMS measurements of the CIS-PB sample** and CIS-RW (even though the latter case is superfluous, given the SEM/EDS analysis supplied at point (26) above. On the other hand, we miss the objection about the SIMS profiles of CIS-RB, as they do appear in Fig. 4a (now Fig. 5).

(28)* *page 6: "The decrease of Cu content in CIS-RB does not imply the formation of a sub-stoichiometric Cu surface, as shown by the Cu/In SIMS ratio (inset)." -> This is not clear to me. I see that the Cu/In ratio decreases at the surface to the level of the bulk of CIS-PW. Since the bulk of PW should be Cu-poor, I would conclude that the surface of RB is also Cu-poor.*

The reviewer is totally correct. The data suggests the formation of a sub-stoichiometric Cu surface in CIS-RB. The text is changed accordingly.

(29)* *page 7: "... the formation of vacancies in multinary chalcogenide materials can be a spontaneous process..." -> You see a reaction when you offer O2 as a reactant. Can this really be called "spontaneous"?*

We propose the following changes to the text: *"This study reveals that the formation of vacancies in multinary chalcogenide materials can be a spontaneously-driven process even at room temperature. The process is enabled by oxidation of the material near the phase boundary."*

Reviewer #3:

The authors studied the relationship between the surface chemistry of CuInSe2 and its device characteristics using photoluminescence spectra, impedance analysis, photo-electrochemical responses under its different surface states. The conclusion that Se deficiency in CuInSe2 is a problem on Voc deficit is reasonable. I partially understood the author's explanations while it needs to overcome some of the issues to be addressed before the publication.

(1) *The authors concluded that photoluminescence from the deep level (0.9 eV) shown in Fig.2(c) originates from indium vacancy, V_{In}, and explained the reason using the chemical reaction formula (04). I think this interpretation is incorrect. According to the theoretical calculation of the defect formation energy using cluster structure (<https://doi.org/10.1016/j.jpcs.2005.09.067>), the formation of V_{In} is hard to occur. Actually, In-Se chemical bonds form the main backbone of the chalcopyrite crystal structure while Cu is easy to migrate. Thus, nobody directly observed indium vacancy. Nevertheless, the authors insist high density of V_{In} introduces 0.9 eV luminescence peak. (I cannot trust it.) If the authors want to claim that indium vacancy originates luminescence peak, then some experimental evidence is required, such as detection it by using positron annihilation. I guess it is possibly caused by In on Cu antisite, In_{Cu}, related defect.*

We thank the referee for raising this issue and **we kindly refer him/her to point (6) of reviewer #2.** Unfortunately, positron annihilation is not a standard routine and we could not get timely access to it. However, in the meantime a new study has been published giving experimental evidence in support to our statement [10]. We agree that our experimental findings seem at odds with the theoretical DFT studies on defect formation energies, including the study published in JPCS. However, **DFT calculations are performed on ideal bulk material, while the formation of In vacancies near the surface involves energetic effects that are not typically taken into account in DFT studies, as now explained.** These considerations are taken into account in the discussion with a comparison to the DFT literature.

(2) *Why did the author not show the result of stoichiometric CuInSe₂, CIS-S, in Fig 3 and 4 (for instance with the depth profile of Se suggested in Fig. 4(a))?* It is well known that the properties of Cu-rich CuInSe₂, CIS-R, are different from those of Cu-poor CuInSe₂, CIS-P, due to its phase diagram. CIS-R must include many Cu₂Se components in the films initially (before KCN etching). I guess the property of CIS-S is somewhat resemble that of CIS-P. In contrast, Cu vacancy, V_{Cu}, concentration affects migration of Na from soda lime glass. (Indeed, the point of Na concentration was not discussed in the manuscript. This is strange.)

Thus the results on CIS-S must be helpful to understand the underlying physics in detail.

We share the view of the referee. However, attaining films with stoichiometric composition requires an extremely laborious experimental effort of finetuning. Therefore, the quantity of material available for further measurements and experiments is limited. Nevertheless, **we have addressed this comment by performing additional SIMS and APT analyses of CIS-SB, which together with the point (27) raised by reviewer#2 provide a compelling series of experiments.** Understandably, the referee points to the absence of a discussion pertaining the effects induced by Na doping. We believe that including such a discussion, however interesting, lies well beyond the scope of the present research. We suggest that a dedicated study incorporating additional Na doping experiments would be more appropriate.

(3) *I agree with the author's suggested model (in Fig.4(b)) that (i) V_{Se} is firstly passivated by air (O₂ or H₂O?) and (ii) form Cu₂Se+In₂O₃ by introducing excess oxygen. But I want to ask the authors why V_{Se} did mainly cause Voc deficit. Actually, V_{Se} is located at 80 meV from conduction band minimum (shallow donor). It may act as self-compensation center. But since the level is too shallow, I think its impact on Voc is not so big. Moreover, V_{Se} is easy to combine with V_{Cu} to form di-vacancy (<https://doi.org/10.1016/j.tsf.2016.02.057>), and many papers suggested the possibility of formation of di-vacancy, which energy level is located deeper in the gap. Why did the authors exclude the discussion on di-vacancy?*

We are very grateful to the referee for pointing us to the cited study. The study provides interesting insights on the formation of divacancy defect complexes. **Indeed, the Cu and Se depletion observed in the near surface region of CIS-RB seems consistent with the formation of divacancy complexes responsible for larger losses in Voc than expected from selenium vacancies alone.** The model explaining the superior performance of CIS-PW and CIS-PB compared to CIS-RB is now strengthened. We suggested some changes to the manuscript to include a short discussion in this regard: a new dedicated figure, as discussed for point (3) of Reviewer # 1, giving reference to this relevant study [15]. The expanded discussion should also address further objection (1) of reviewer#1 on Figure 4b.

(4) *Why did not the author mention the efficiency of the KCN etching on CIS-R to remove Cu₂Se? Referring to Fig. 1(b), it seems the CIS-R seems to have been grown out of the CIS metastable region and in the Cu₂Se + CIS region. It is moreover shown later in the article that a weak KCN etching can leave a significant amount of Cu₂Se on the sample, which has a strong effect on its properties. I think it would be interesting for the discussion to include the amount of Cu₂Se (if any) remaining after the KCN etching on samples presented at the beginning of the article.*

This comment is in line with point (26) of reviewer#2. **We now supply the corresponding evidence of incomplete removal of the Cu-Se phases from the surface of CIS-RW acquired by SEM/EDS.** Figure 1b shows the microstructure of the Cu-rich CIS film after KCN etching, which we think are more relevant for the discussion at that point in the manuscript. However, the text is now part of a separate section and has been heavily edited, so that the point made should now be clear. Images of a Cu-rich film after weak and no etching are now also supplied as a new figure in the manuscript.

References

- [1] D. Colombara, A.-M. Gonçalves, and A. Etcheberry, "Synthesis of K₂Se solar cell dopant in liquid NH₃ by solvated electron transfer to elemental selenium," *Electrochemistry Communications*, vol. 93, pp. 44–48, 2018.
- [2] D. Colombara, P. J. Dale, G. P. Kissling, L. M. Peter, and S. Tombolato, "Photoelectrochemical Screening of Solar Cell Absorber Layers: Electron Transfer Kinetics and Surface Stabilization," *J. Phys. Chem. C*, vol. 120, no. 29, pp. 15956–15965, Apr. 2016.
- [3] S. Lany and A. Zunger, "Light- and bias-induced metastabilities in Cu(In,Ga)Se₂ based solar cells caused by the (VSe-VCu) vacancy complex," *Journal of Applied Physics*, vol. 100, no. 11, p. 113725, Dec. 2006.
- [4] E. Avancini *et al.*, "Impact of compositional grading and overall Cu deficiency on the near-infrared response in Cu(In, Ga)Se₂ solar cells," *Prog. Photovolt: Res. Appl.*, p. n/a-n/a, Jan. 2016.
- [5] P. Jackson, D. Hariskos, R. Wuerz, W. Wischmann, and M. Powalla, "Compositional investigation of potassium doped Cu(In,Ga)Se₂ solar cells with efficiencies up to 20.8%," *physica status solidi (RRL) – Rapid Research Letters*, vol. 8, no. 3, pp. 219–222, 2014.
- [6] Z. Yuan *et al.*, "Na-Diffusion Enhanced p-type Conductivity in Cu(In,Ga)Se₂: A New Mechanism for Efficient Doping in Semiconductors," *Advanced Energy Materials*, vol. 6, no. 24, p. 1601191, Dec. 2016.
- [7] L. E. Oikkonen, M. G. Ganchenkova, A. P. Seitsonen, and R. M. Nieminen, "Effect of sodium incorporation into CuInSe₂ from first principles," *Journal of Applied Physics*, vol. 114, no. 8, p. 083503, Aug. 2013.
- [8] D. Colombara, "Frank-Turnbull dopant migration may enhance heteroatom diffusivity: Evidence from alkali-doped Cu(In,Ga)Se₂," *Phys. Rev. Materials*, vol. 3, no. 5, p. 054602, May 2019.
- [9] B. M. Kowalski, N. Manz, D. Bethke, E. A. Shaner, A. Serov, and N. G. Kalugin, "Role of humidity in oxidation of ultrathin GaSe," *Mater. Res. Express*, 2019.
- [10] F. Babbe, H. Elanzeery, M. H. Wolter, K. Santhosh, and S. Siebentritt, "The hunt for the third acceptor in CuInSe₂ and Cu(In,Ga)Se₂ absorber layers," *Journal of Physics: Condensed Matter*, vol. 31, p. 425702, 2019.
- [11] E. Simsek Sanli *et al.*, "Point defect segregation and its role in the detrimental nature of Frank partials in Cu(In,Ga)Se₂ thin-film absorbers," *Phys. Rev. B*, vol. 95, no. 19, p. 195209, May 2017.
- [12] Y. Li *et al.*, "Segregation Stabilizes Nanocrystalline Bulk Steel with Near Theoretical Strength," *Phys. Rev. Lett.*, vol. 113, no. 10, p. 106104, Sep. 2014.
- [13] J. J. Scragg, P. J. Dale, D. Colombara, and L. M. Peter, "Thermodynamic Aspects of the Synthesis of Thin-Film Materials for Solar Cells," *ChemPhysChem*, vol. 13, no. 12, pp. 3035–3046, 2012.
- [14] M. Jošt *et al.*, "21.6%-Efficient Monolithic Perovskite/Cu(In,Ga)Se₂ Tandem Solar Cells with Thin Conformal Hole Transport Layers for Integration on Rough Bottom Cell Surfaces," *ACS Energy Lett.*, vol. 4, no. 2, pp. 583–590, Feb. 2019.
- [15] A. Uedono *et al.*, "Vacancy behavior in Cu(In_{1-x}Ga_x)Se₂ layers grown by a three-stage coevaporation process probed by monoenergetic positron beams," *Thin Solid Films*, vol. 603, pp. 418–423, Mar. 2016.

Reviewers' comments:

Reviewer #1 (Remarks to the Author):

The authors have addressed some of the concerns of the review and massively changed the manuscript. However, despite carefully going through the manuscript several times, I am left confused at various points of the manuscript and have difficulties understanding and interpreting what the authors want to say.

I would say the additional figures and discussion did not solely improve the readability, and it seems that the message has become even less focused. The figures are now very crowded and far from self explanatory. The acronyms are very confusing, it is almost impossible to keep track what is being discussed at a given point.

Detailed comments are below:

(1) New Title: "Chemical instability of chalcogenide surfaces : technological impact well beyond the surface" ?

- The authors changed the title but I cannot see that it works; why chalcogenide surfaces – they talk very specifically about CuInSe₂ surfaces and phases in the manuscript. Also, what is the technological impact of this ? Reading through the paper careful several times, I could not find any statement related to such an impact, or which would justify this phrasing in the title

(2) I still do not understand the section on SPV and PEC and in particular the lifetimes deduced (way too long) and the interpretation of bulk versus interface defects from this.

(3) Defects that may be formed. The authors now conclude that the surface treatment generates Lany-Zunger type defects, and in the discussion in the end even imply there would be quantitative relation between the PEC and SPV results and the theory of these defects. I do not see how the authors can make such a bold statement.

DFT theory also shows that selenium vacancies and V_{Se}-V_{Cu} defect pairs have very large formation energies (e.g. see Phys. Rev. 87 (2013) 245203), which would make the formation very unlikely or even impossible in particular at room temperature. This is ignored in the manuscript.

(4) "It is hypothesized that the formation energy of V_{In} in CIS (ca. 2.5 eV 57,58) resulting from the reaction

between indium atoms and gas-phase oxygen is offset energetically by the large Gibbs free energy gain associated to (O₂) (ca. 2.9 eV/mol)."

- How does the formation energy per defect compare with free energy gain of eV/mol ?

-

(5) Solar cell JV curves are given but no efficiencies. If the discussion on the device performance is relevant then this should be added

(6) Figure 2(a) is way too small to be read;

2b and 2c: The Se-content from APT of 45% is extending into the bulk up to 300nm (this figure is new); If this is quantitative it should be explained how the Se content can be so low in CIS or CIS and secondary phases, or the assumption is there are 5% Selenium vacancies ?

It is not clear what the Figures 2b and 2c are about, and why there are 2 of them, and why the axis are flipped on them.

(7) „Cu Se phases are very conductive and reduce severely the shunt resistance of chalcogenide -

based solar cells"

-I think the authors mean chalcopyrite or Cu-chalcogenide; e.g, CdTe is a chalcogenide solar cell but does not suffer from CuSe

(8) The histogram in Fig. 6c shows the relative activation energy (E_a) of the main capacitance step extracted from the slope of the Arrhenius plot of the ADM spectra

-This is not a histogram

(9) CuInSe₂-O₂ thermochemistry and metastability:

What do the authors mean by this metastability ? In the paragraph metastability is only mentioned in conjunction with metastable Se₂ molecules and in the final statement : "It is then revealed that interface reactions can lead to a chalcogenide material with metastable defect populations at temperatures as high as 550C."

-What do the authors really want to say about metastability? This is not clear throughout the manuscript

(10) The acronyms (still) do not work. I give some examples of incomprehensible sentences:

"However, it is clear that RB is affected by a significant deficit of open circuit voltage (VOC) compared to both PW and PB. In light of the PEC and ADM Schottky analysis, KCN-B etching could then be responsible for the formation of defects that cause interface recombination at the RB/buffer interface, hence for the enduring VOC deficit of CIS-R compared to CIS-P"

".. for CIS-R films, CIS-RB luminesces more than CIS-RW"

"This is attributed to the fact the KCN-W is unable to remove the...,as it is most evident for the CIS-RU"

It is not possible to keep track of the acronyms and read through the text knowing what a specific section is about.

(11) caption figure 8: the lower degree of band flattening (dashed lines) compared to the Cu-poor case, as observed by TR-SPV, and (II) the slowly increasing photocurrent transient, instead of the decreasing photocurrent typical of Cu-poor CIS, as observed by PEC

-I do not understand this sentence

(12) "entails a sizable activation energy for the cleavage of the acceptor via capture of two holes, hence the time constant associated to the surface charge reconstruction is on the order of tens of minutes at room temperature, as observed in TR-SPV."

How is the relevant time scale of tens of minutes quantitatively estimated ?

(13) Figure 3(a) cannot be read

Reviewer #2 (Remarks to the Author):

The authors have addressed in much detail the questions and concerns raised by the reviewers and added additional explanations and experimental evidence to the manuscripts. The manuscripts has certainly gained clarity, although it remains a paper that is challenging to read.

In general, I think this paper should be published as soon as possible. However some remaining points should be considered:

* You claim that Fig. 4a reveals Cu-Se outgrowth. However I don't see that this claim can be derived from the Figure. For this a reference of non-exposed CIS-R would be needed. I understand if this is out of scope of this paper, but then the wording (using "reveal") is not appropriate.

* PEC: If I understand it right, you explain the decrease of the current for PB by the accumulation of carriers near the interface. Do you really expect that the electron density increases over a time-span of 50 μ s? This would require a huge carrier life-time, right? Also I don't see how this accumulation would decrease the current density since more carriers would be available for a charge transfer to the electrolyte. I must admit that I didn't read the cited work by Peter, but I wonder if the slow decrease of the current isn't rather due to a depletion of Eu^{3+} at the interface due to slow diffusion of the electrolyte. Besides this, could the change of the PEC current also be due to an alteration of the surface chemistry caused by the electrolyte during illumination?

* PDT: I guess that the samples were heated during PDT treatments, right? (Please provide this information in the Methods). Can you exclude that the changes you see by PDT are caused by the heating rather than the incorporation of Se? Did you heat a sample without deposition of Se?

* V_{In} defect: You say that the high barrier for In migration leads to persistence of V_{In} . But shouldn't this also prevent out-diffusion for In_2O_3 formation in the first place?

Minor remarks:

* I still find some of the graphs too small with font sizes and colors that are impossible to read on a printout. For example Fig. 6b would be easier to understand if you give it more space on the horizontal dimension. The red and blue labels on grey background in Fig. 4b and c have too little contrast and/or are too small to read.

* Do you need ROI 1 in Fig. 2a? I couldn't see that it is used anywhere.

* In Fig 5b: Are the differences for KCN-W and KCN-B at 7kV significant? Can you give uncertainties for EDS?

Reviewer #3 (Remarks to the Author):

The authors are responding to my peer-reviewed comments and no further modifications to my comments are necessary. However, the authors need to respond seriously to the comments of the other reviewers (my impression is close to 1st reviewer).

COMMENTS TO AUTHOR:

Reviewer #1:

The authors have addressed some of the concerns of the review and massively changed the manuscript. However, despite carefully going through the manuscript several times, I am left confused at various points of the manuscript and have difficulties understanding and interpreting what the authors want to say. I would say the additional figures and discussion did not solely improve the readability, and it seems that the message has become even less focused. The figures are now very crowded and far from self explanatory. The acronyms are very confusing, it is almost impossible to keep track what is being discussed at a given point.

We divided the manuscript in manageable sections in order to guide the reader and ease understanding. Additionally, we have now **(i) moved the least central parts of the manuscript to the supplementary information in order to streamline the argument's flow and highlight the scientific message, which also simplified figure 2 and 3; (ii) disambiguated the lexicon and (iii) heavily decreased the use of acronyms within the text while also including a detailed table with sample naming at the beginning of the article.**

(1) New Title: "Chemical instability of chalcogenide surfaces : technological impact well beyond the surface". - The authors changed the title but I cannot see that it works; why chalcogenide surfaces – they talk very specifically about CuInSe₂ surfaces and phases in the manuscript. Also, what is the technological impact of this ? Reading through the paper careful several times, I could not find any statement related to such an impact, or which would justify this phrasing in the title.

We have changed the title to the following: **"Chemical instability of chalcogenide surfaces: technological impact on chalcopyrite well beyond the surface"**. The approach presented here for chalcopyrite has general implications for other materials, even beyond chalcogenides; however, since the focus is on chalcopyrite, the new title is more appropriate. As per the technological impact, the title refers to the effect of the modified surfaces on device performance clearly presented at the end of the manuscript through a dedicated table (Table 1).

(2) I still do not understand the section on SPV and PEC and in particular the lifetimes deduced (way too long) and the interpretation of bulk versus interface defects from this.

We understand the comment pointing out a potential confusion of the reader, because these techniques are not very frequently used and commonly known. A similar concern is raised by Reviewer#2 (see comment 2). **It is crucial to understand that the time constants determined by PEC and SPV and discussed in this section are NOT carrier life times, but time constants of electron transfer and of defect transformation/generation.** We ensure to never use the term "lifetime", as we speak about time constants.

The term "surface recombination" is used in the electrochemical literature to imply electron transfer to leaky surface states, it would be inappropriate to invent a new term. As discussed previously, the PEC electron transfer kinetics depends on the extent of energetic overlap between the states in the band and the distribution of redox states in solution, and on the mechanism of electron transfer to the oxidised species in solution (i.e. outer-sphere or inner-sphere [1]). **Typical rate constants are on the order of 10^2 s^{-1} . So, the time constants we observe are perfectly normal and are those related to the electron transfer not those of the actual recombination process.** We modified the text to clarify this fact. Within the PEC lexicon, the word recombination is intended as "surface recombination" and typically refers to phenomena happening at the semiconductor surface occurring exclusively during PEC experiments, so no comparison to SPV can be made (nor it is claimed).

However, our PEC findings reveal for the first time **an additional signature** in the photocurrent transient that we attribute to a phenomenon occurring in the bulk of CIS (as explained by comparison to SPV analyses and DFT calculations). Normally, charge redistribution phenomena happening in the bulk of semiconductors are not resolved by PEC because their time constant is far too short compared to the acquisition rate capabilities of standard potentiostats. However, in this specific case, the time

constant associated to defect formation in the bulk (or near the surface) of Cu-rich CIS subject to bold KCN etching happens to be so exceptionally long that it yields a sizable time domain contribution in the PEC transient. It is by relating this effect to the SPV transients that we were able to identify its common solid-state nature! The time constant we observe is not the carrier lifetime but the time needed for the defect, involved in the minority carrier recombination, to change its charge state.

The intensity-modulated photocurrent spectroscopy (IMPS) measurements were performed to quantify the time constants associated with the charge redistribution phenomena revealed by the odd shape of the PEC photocurrent transient. This is done by analysis of the frequency domain, i.e. by identifying the frequency at which the imaginary photocurrent reaches its maximum [2]; that frequency corresponds to the time constant of electron transport. A time constant of ca. 2 microseconds is identified for both Cu-poor and Cu-rich films subject to bold KCN etching, **which is in line with the value determined from SPV for Cu-rich films subject to bold KCN etching**. The extent of “surface recombination” in Cu-rich CIS after bold KCN (RB) is ca. one order of magnitude higher than in Cu-poor (PB), in line with the PEC results (Fig. 6a inset). The lower photocurrent in RB is consistent with a higher extent of surface recombination; however, we cannot exclude that the effect is due to a lower mobility. **Most importantly, these experimental findings corroborate a theoretical model published by Lany and Zunger [3]** (S. Lany and A. Zunger, “Light- and bias-induced metastabilities in Cu(In,Ga)Se₂ based solar cells caused by the (V_{Se}-V_{Cu}) vacancy complex,” *Journal of Applied Physics*, vol. 100, no. 11, p. 113725, Dec. 2006), **as clearly described in the dedicated section**.

(3) Defects that may be formed. The authors now conclude that the surface treatment generates Lany-Zunger type defects, and in the discussion in the end even imply there would be quantitative relation between the PEC and SPV results and the theory of these defects. I do not see how the authors can make such a bold statement. DFT theory also shows that selenium vacancies and V_{Se}-V_{Cu} defect pairs have very large formation energies (e.g. see Phys. Rev. 87 (2013) 245203), which would make the formation very unlikely or even impossible in particular at room temperature. This is ignored in the manuscript.

There is no such claim of *quantitative* relation between PEC/SPV results and DFT analysis by Lany and Zunger. We have just quantified the time constants of the processes recorded in both PEC and SPV analyses and concluded that such long time constants are consistent (i.e. trend going in the same direction) with the DFT model by Lany and Zunger. As just the title of Lany and Zunger’s ref. [3] reveals, their model is designed *precisely* to address theoretically the light-induced metastabilities that were widely observed experimentally by other means.

The newer DFT results by Pohl and Albe do not exclude the explanation of metastable behaviour by the Lany-Zunger model. They show, that in **thermal equilibrium** the formation of the Se vacancy is not favourable, but neither growth nor our etching process are equilibrium processes. We do leach Se from the absorber surface, as we have shown in ref. [4]. Furthermore, the conditions during etching are likely close to point D in the phase diagram of Pohl and Albe, where the formation energy of the Se vacancy is lowest. This is not to be confused with growth conditions, which are close to point A (for Cu-poor) and B (for Cu-rich). We remove Se from the crystal during the etching process. Thus, the crystal has to form Se vacancies – what else should happen? There is experimental evidence that even in Cu-rich CuInSe₂ Cu vacancies exist (for a review see ref. [5], C Spindler, F Babbe, MH Wolter, F Ehré, K Santhosh, P Hilgert, F Werner, and S Siebentritt, "Electronic Defects in Cu(In,Ga)Se₂ – towards a comprehensive model" *Physical Review Materials* **3**, 090302 (2019)). Thus, it is very reasonable to assume that the double vacancy forms. The metastable behaviour of the Cu-Se double vacancy has been confirmed by Pohl and Albe and is still the best explanation we have for the many metastable effects observed in Cu(In,Ga)Se₂ solar cells. It has been used already 4 times in papers appearing in 2020 to explain metastable effects (refs. [6]–[9]), strengthening further the timeliness of our study. We added the discussion of the new DFT results in the supporting information.

(4) “It is hypothesized that the formation energy of V_{In} in CIS (ca. 2.5 eV [57,58]) resulting from the reaction between indium atoms and gas-phase oxygen is offset energetically by the large Gibbs free energy gain associated to (O₂) (ca. 2.9 eV/mol).”- How does the formation energy per defect compare with free energy gain of eV/mol ?

We apologize for erroneously reporting the Gibbs free energy gain associated to (02) as ca. 2.9 eV/mol instead of 2.9 eV per In atom (i.e. 550 kJ/mol of In_2O_3 corresponds to $1.72 \cdot 10^{24}$ eV/mol of In atom).

As a side note, it is now confirmed that the shoulder peak in the PL spectra is due to V_{In}. For a detailed explanation we direct the reviewer to ref. [10] (Babbe, F., Elanzeery, H., Wolter, M. H., Santhosh, K. & Siebentritt, S. The hunt for the third acceptor in CuInSe₂ and Cu(In,Ga)Se₂ absorber layers. Journal of Physics: Condensed Matter 31 425702 (2019)) and ref. [5] (C Spindler, F Babbe, MH Wolter, F Ehré, K Santhosh, P Hilgert, F Werner, and S Siebentritt, "Electronic Defects in Cu(In,Ga)Se₂ – towards a comprehensive model" Physical Review Materials 3, 090302 (2019)).

(5) *Solar cell JV curves are given but no efficiencies. If the discussion on the device performance is relevant then this should be added*

We agree with the reviewer. These are now added in a dedicated table:

Sample	Efficiency / %	V _{oc} / mV	J _{sc} / mAcm ⁻²	F.F. / %
PW [11]	12.8	446	42.0	68
PB	12.7	460.2	44.5	61.9
SW [12]	12.1	436.8	40.7	67.9
SB	11.7	434.5	38.5	69.7
RW [4]	-	-	-	-
RB [4]	7.0	355	42.1	46.8
RB+Se only [4]	9.2	382	40.7	59.2
RB+Se KF PDT [13]	9.5	398	41.1	58.0
RB+Zn(O,S) [4]	6.7	399	41.0	40.9

(6) *Figure 2(a) is way too small to be read; 2b and 2c: The Se-content from APT of 45% is extending into the bulk up to 300nm (this figure is new); If this is quantitative it should be explained how the Se content can be so low in CIS or CIS and secondary phases, or the assumption is there are 5% Selenium vacancies? It is not clear what the Figures 2b and 2c are about, and why there are 2 of them, and why the axis are flipped on them.*

The figure has been rearranged, so that 2a is now larger. 2b and 2c refer to the two regions of interest labelled in 2a. The axis of 2b is now flipped to avoid confusion when comparing 2b and 2c. The quantification in APT clearly reveals Se deficiency with respect to stoichiometry. However, we do not claim that the whole Se deficiency is translated into Se vacancies, as vacancies cannot directly be measured by APT technique. Partly, the Se deficiency might be due to a preferential loss of Se during APT measurement, which is a well-known issue for semiconductor compounds [14]–[16]. Partly, it may be consistent with the hypothesis of interspersed Cu₂Se domains.

(7) *„Cu Se phases are very conductive and reduce severely the shunt resistance of chalcogenide - based solar cells” -I think the authors mean chalcopyrite or Cu-chalcogenide; e.g, CdTe is a chalcogenide solar cell but does not suffer from CuSe*

We agree with the reviewer, the text is modified accordingly.

(8) *The histogram in Fig. 6c shows the relative activation energy (E_a) of the main capacitance step extracted from the slope of the Arrhenius plot of the ADM spectra -This is not a histogram*

We agree with the reviewer, we now call it bar chart.

(9) *CuInSe₂-O₂ thermochemistry and metastability: What do the authors mean by this metastability? In the paragraph metastability is only mentioned in conjunction with metastable Se₂ molecules and in the final statement: "It is then revealed that interface reactions can lead to a chalcogenide material with metastable defect populations at temperatures as high as 550C." -What do the authors really want to say about metastability? This is not clear throughout the manuscript*

We understand that this is a source of misunderstanding.

Metastability is a term used by different communities entailing essentially the same concept but applied to different phenomena. In this manuscript it is employed both for phase equilibria and electronic defects. Sometimes there is conceptual overlap as well, which justifies the wording to encompass both concepts at once, sometimes it only refers to phase equilibria, sometimes only to the light-induced metastable effects. **In order to clarify this point right at the beginning, the following sentence is included at the end of the introduction:**

"Throughout the manuscript the word metastability refers to a sample condition describing a stable state of a dynamical system other than the system's state of least energy. Depending on the context, the term applies both to phase equilibria as well as electronic defect states."

(10) *The acronyms (still) do not work. I give some examples of incomprehensible sentences: "However, it is clear that RB is affected by a significant deficit of open circuit voltage (VOC) compared to both PW and PB. In light of the PEC and ADM Schottky analysis, KCN-B etching could then be responsible for the formation of defects that cause interface recombination at the RB/buffer interface, hence for the enduring VOC deficit of CIS-R compared to CIS-P" ".. for CIS-R films, CIS-RB luminesces more than CIS-RW" "This is attributed to the fact the KCN-W is unable to remove the..., as it is most evident for the CIS-RU" It is not possible to keep track of the acronyms and read through the text knowing what a specific section is about.?*

We understand the referee's frustration. We find it very hard to identify a simpler system for sample naming. We have now opted for a substantial reduction of acronyms throughout the text. Although this leads to a slight lengthening of the sentences, we hope that the text has now gained clarity. Due to space restriction, the acronyms are still included within the figures, but spelled out entirely within each figure caption. The new Table 1 should also ease the reader in this regard.

(11) *caption figure 8: the lower degree of band flattening (dashed lines) compared to the Cu-poor case, as observed by TR-SPV, and (II) the slowly increasing photocurrent transient, instead of the decreasing photocurrent typical of Cu-poor CIS, as observed by PEC -I do not understand this sentence.*

Indeed, this sentence is confusing because it refers to two cases (Cu-rich and Cu-poor), while the figure only depicts one case. Figure and caption have been modified to improve clarity.

(12) *"entails a sizable activation energy for the cleavage of the acceptor via capture of two holes, hence the time constant associated to the surface charge reconstruction is on the order of tens of minutes at room temperature, as observed in TR-SPV." How is the relevant time scale of tens of minutes quantitatively estimated?*

There is clearly no need for precise quantification of this time constant. The SPV data of Cu-rich CIS subject to bold KCN etching (RB) in Fig. 6b clearly shows that after more than 17 minutes, the surface photovoltage has still not recovered the initial in-dark value. It is beyond the objective of this study to provide a quantitative estimation of such long time constants.

(13) *Figure 3(a) cannot be read*

We agree with the reviewer. Following our attempt of streamlining the manuscript's flow, the inset has been moved to the supplementary information, so Fig. 3a is now fully readable.

Reviewer #2:

The authors have addressed in much detail the questions and concerns raised by the reviewers and added additional explanations and experimental evidence to the manuscripts. The manuscripts has certainly gained clarity, although it remains a paper that is challenging to read. In general, I think this paper should be published as soon as possible.

We thank the referee for the support and for the additional comments raised.

However some remaining points should be considered:

*(1) * You claim that Fig. 4a reveals Cu-Se outgrowth. However I don't see that this claim can be derived from the Figure. For this a reference of non-exposed CIS-R would be needed. I understand if this is out of scope of this paper, but then the wording (using "reveal") is not appropriate.*

We understand the referee's point. Unfortunately, we are not equipped to transfer samples freshly obtained by co-evaporation or after etching into the XPS chamber without contact to air. Since XPS is an extremely surface-sensitive technique, it is virtually impossible for us to obtain the XPS signature of pristine (i.e. non-exposed) samples. Therefore, we have softened the claim stating that XPS reveals a higher concentration of Cu and Se, which is compatible with a larger Cu-Se outgrowth. We believe that all other data sets presented are complete.

*(2) * PEC: If I understand it right, you explain the decrease of the current for PB by the accumulation of carriers near the interface. Do you really expect that the electron density increases over a time-span of 50 μ s? This would require a huge carrier life-time, right? Also I don't see how this accumulation would decrease the current density since more carriers would be available for a charge transfer to the electrolyte. I must admit that I didn't read the cited work by Peter, but I wonder if the slow decrease of the current isn't rather due to a depletion of Eu^{3+} at the interface due to slow diffusion of the electrolyte. Besides this, could the change of the PEC current also be due to an alteration of the surface chemistry caused by the electrolyte during illumination?*

We would like to highlight that the photocurrent transients in the PEC experiment are recorded after a light pulse, but the light pulse lasts hundreds of milliseconds; during this time span the sample is subject to constant illumination and will reach a steady-state equilibrium between the photogenerated electron flux on one hand and the sum of electron fluxes corresponding to the electrons transferred to the oxidised species in solution plus the electrons filling the leaky surface states. The resulting time constant is a trade-off between the rate constants for electron transfer to the electrolyte (which was measured for CIS/ Eu^{3+} in a separate study at the value of 250 s^{-1} [2]) and the rate constant for the surface recombination phenomenon (which is thus estimated at $2\div 4 \text{ s}^{-1}$). So the carrier lifetimes do not need to be huge for this effect to be observed, they just need to be sufficiently long to allow electron diffusion from the depth at which they are generated to the semiconductor surface. It is stressed that the nature of the surface states being filled is unknown. It is not an objective of the present study to unravel its origin, although the correlation to the XPS results suggests strongly that it could be related to the amount of conductive Cu-Se phases forming at the surface of the CIS films, the rate constant being higher for RW than for PW, and increasing in both cases upon exposure of the films to air, i.e. when reaction (O2) is likely taking place.

As per the electrolyte diffusion, this would be an issue at much lower concentration of electrolyte; at 0.2 molar concentration such effects are negligible. In any case, what is striking is the consistently different behavior between Cu-poor and Cu-rich material, which cannot depend on diffusion effects in the electrolyte, since the same solution was used in the two cases. Lastly, we exclude any long-term effect induced by the electrolyte during illumination. Several on/off cycles are typically measured for each dataset, e.g. for each point in the inset of Fig. 6a. Reproducible transients are recorded within each set of measurements, if we exclude the effect of surface oxidation caused by the exposure of the bare absorbers to air between different sets of measurements.

The figure reported here shows a typical dataset in which several on/off cycles are recorded, yielding largely reproducible transients. In this specific case the black solid line corresponds to an aged Cu-poor CIS, while the dashed red line corresponds to the same sample after KCN etching (5 wt. %, 30 seconds). KCN is known to selectively remove Cu-Se conductive phases from the surface of CIS, which leads to a large boost of electrochemical photocurrent (i.e. a decrease of the rate constant for surface recombination).

(3) * PDT: I guess that the samples were heated during PDT treatments, right? (Please provide this information in the Methods). Can you exclude that the changes you see by PDT are caused by the heating rather than the incorporation of Se? Did you heat a sample without deposition of Se?

Indeed, the heating effect was studied on Cu-rich CIS absorbers. Heating the absorbers at temperatures similar to the Se-treatment but with no Se incorporation lead to the deterioration of the formed CIS phase causing a significant decrease in the V_{OC} , FF and consequently the efficiency. Only after incorporating the Se, the CIS phase is maintained. Therefore, we can confirm that the changes observed by the PDT are related to the Se incorporation and not purely to the heating effect.

(4) * V_{In} defect: You say that the high barrier for In migration leads to persistence of V_{In} . But shouldn't this also prevent out-diffusion for In_2O_3 formation in the first place?

This is an intriguing question. The migration barriers calculated by DFT correspond to bulk unit cells. It is very likely that such values do not apply at the surface of a material. V_{In} may then form because the outdiffusion near the surface may be less energetically demanding. However, when V_{In} are formed near the surface of CIS, In atoms from the bulk of the material will have harder time migrating.

(5) * I still find some of the graphs too small with font sizes and colors that are impossible to read on a printout. For example Fig. 6b would be easier to understand if you give it more space on the horizontal dimension. The red and blue labels on grey background in Fig. 4b and c have too little contrast and/or are too small to read.

Thank you. We have fixed these points. The abscissa in Fig. 6b has been broken, in order to have more space for the SPV signal under the illumination sequence. The labelling colour in Fig. 4b has been changed to improve the contrast in the figure.

(6) * Do you need ROI 1 in Fig. 2a? I couldn't see that it is used anywhere.

ROI 1 is used in Fig. 2b. However, the figure has been improved also to accommodate comment #6 of Reviewer#1.

(7) * In Fig 5b: Are the differences for KCN-W and KCN-B at 7kV significant? Can you give uncertainties for EDS?

The table below reports the average values obtained by EDS on multiple areas of the films. The differences of Cu/In and (Cu+In)/Se ratios appear to be meaningless when comparing the values corresponding to CIS-P between KCN-W and KCN-B. The differences are rather meaningless for CIS-S, but meaningful for CIS-R.

		KCN-W	KCN-B
CIS-P	Cu/In	0.850±0.008	0.850±0.007
	(Cu+In)/Se	1.125±0.012	1.126±0.001
CIS-S	Cu/In	0.944±0.010	0.935±0.010
	(Cu+In)/Se	1.135±0.007	1.148±0.008
CIS-R	Cu/In	0.897±0.062	0.768±0.013
	(Cu+In)/Se	1.414±0.036	1.308±0.017

Reviewer #3:

The authors are responding to my peer-reviewed comments and no further modifications to my comments are necessary. However, the authors need to respond seriously to the comments of the other reviewers (my impression is close to 1st reviewer).

We thank the referee for the support and hope that our response to the other two referees addresses appropriately their concerns.

References

- [1] D. Colombara, A.-M. Gonçalves, and A. Etcheberry, "Synthesis of K₂Se solar cell dopant in liquid NH₃ by solvated electron transfer to elemental selenium," *Electrochemistry Communications*, vol. 93, pp. 44–48, 2018, doi: 10.1016/j.elecom.2018.06.002.
- [2] D. Colombara, P. J. Dale, G. P. Kissling, L. M. Peter, and S. Tomblato, "Photoelectrochemical Screening of Solar Cell Absorber Layers: Electron Transfer Kinetics and Surface Stabilization," *J. Phys. Chem. C*, vol. 120, no. 29, pp. 15956–15965, Apr. 2016, doi: 10.1021/acs.jpcc.5b12531.
- [3] S. Lany and A. Zunger, "Light- and bias-induced metastabilities in Cu(In,Ga)Se₂ based solar cells caused by the (VSe-VCu) vacancy complex," *Journal of Applied Physics*, vol. 100, no. 11, p. 113725, Dec. 2006, doi: 10.1063/1.2388256.
- [4] H. Elanzeery, M. Melchiorre, F. Babbe, M. Sood, F. Werner, and S. Siebentritt, "Challenge in Cu-rich CuInSe₂ thin film solar cells: Defect caused by etching," *Physical Review Materials*, vol. 3, p. 055403, 2019.
- [5] C. Spindler *et al.*, "Electronic Defects in Cu(In,Ga)Se₂: Towards a Comprehensive Model," *Phys. Rev. Materials*, vol. 3, no. 9, p. 090302, Sep. 2019, doi: 10.1103/PhysRevMaterials.3.090302.
- [6] I. Khatri, T. Yashiro, T.-Y. Lin, M. Sugiyama, and T. Nakada, "Metastable Behavior on Cesium Fluoride-Treated Cu(In_{1-x}Ga_x)Se₂ Solar Cells," *physica status solidi (RRL) – Rapid Research Letters*, vol. 14, no. 4, p. 1900701, 2020, doi: 10.1002/pssr.201900701.
- [7] M. Nardone *et al.*, "Quantifying Large Lattice Relaxations in Photovoltaic Devices," *Phys. Rev. Applied*, vol. 13, no. 2, p. 024025, Feb. 2020, doi: 10.1103/PhysRevApplied.13.024025.
- [8] J. Sastré-Hernández *et al.*, "Systematized and simplified processing of CuInGaSe₂ thin films to be applied on solar cells," *Chalcogenide Letters*, vol. 17, no. 2, pp. 69–76, 2020.
- [9] M. Ballabio, D. F. Marrón, N. Barreau, M. Bonn, and E. Cánovas, "Composition-Dependent Passivation Efficiency at the CdS/CuIn_{1-x}Ga_xSe₂ Interface," *Advanced Materials*, vol. 32, no. 9, p. 1907763, 2020, doi: 10.1002/adma.201907763.
- [10] F. Babbe, H. Elanzeery, M. H. Wolter, K. Santhosh, and S. Siebentritt, "The hunt for the third acceptor in CuInSe₂ and Cu(In,Ga)Se₂ absorber layers," *Journal of Physics: Condensed Matter*, vol. 31, p. 425702, 2019.
- [11] H. Elanzeery, F. Babbe, M. Melchiorre, A. Zelenina, and S. Siebentritt, "Potassium Fluoride Ex Situ Treatment on Both Cu-Rich and Cu-Poor CuInSe₂ Thin Film Solar Cells," *IEEE Journal of Photovoltaics*, vol. 7, no. 2, pp. 684–689, Mar. 2017, doi: 10.1109/JPHOTOV.2017.2651802.
- [12] H. Elanzeery, "The cause of interface recombination in Cu-rich CIS thin film solar cells," University of Luxembourg, Esch-sur-Alzette, Luxembourg, 2019.
- [13] F. Babbe, H. Elanzeery, M. Melchiorre, A. Zelenina, and S. Siebentritt, "Potassium fluoride post deposition treatment with etching step on both Cu rich and Cu poor CuInSe₂ thin film solar cells," *Physical Review Materials*, vol. 2, p. 105405, 2018.
- [14] D. Colombara *et al.*, "Sodium enhances indium-gallium interdiffusion in copper indium gallium diselenide," *Nature Communications*, vol. 9, p. 826, 2018, doi: 10.1038/s41467-018-03115-0.
- [15] M. Müller, D. W. Saxey, G. D. W. Smith, and B. Gault, "Some aspects of the field evaporation behaviour of GaSb," *Ultramicroscopy*, vol. 111, no. 6, pp. 487–492, May 2011, doi: 10.1016/j.ultramic.2010.11.019.
- [16] L. Mancini *et al.*, "Composition of Wide Bandgap Semiconductor Materials and Nanostructures Measured by Atom Probe Tomography and Its Dependence on the Surface Electric Field," *J. Phys. Chem. C*, vol. 118, no. 41, pp. 24136–24151, Oct. 2014, doi: 10.1021/jp5071264.

REVIEWERS' COMMENTS:

Reviewer #1 (Remarks to the Author):

The authors have responded to the raised concerns of the review and greatly improved the readability and consistency of the manuscript. I remain at odds with one point: the title claiming "technological impact" and the statement "Tailored chemical treatments mitigate anion vacancy formation and boost the efficiency of CuInSe₂ solar cells." in the abstract, as well as the statement related to table 1 "the technological impact of the findings I summarised in Table 1, where all the device performance parameters of the various samples are reported"

To my understanding the authors explain in detail phase transformations and defect formation as well as annihilation occurring at the surface of chalcopyrite thin films, for films with compositions close to the edge of the single phase region. With respect to devices or technology, they confirm (and explain) what has been done for Cu(In,Ga)Se₂ device manufacturing for many years, the utilization of Cu-poor compositions for high performance solar cells. To my knowledge they do not have to be KCN etched to yield high performance. The best CuInSe₂ solar cells to my knowledge are close to 14%, somewhat larger than the devices presented in table 1 (not discussed by the authors), and I do not see that the treatments proposed in this study would yield to a "boost in efficiency" of solar cells. In that sense "tailored" seems also not correct, since the best solar cells in table 1 are the PW and PB treatments, two different etchings yielding almost identical efficiencies.

Of course, if the "boost in efficiency" statement is related to the increase in efficiency of Cu-rich grown CuInSe₂ (as I think it is) then this becomes misleading to the reader, since the statement relates to a moderate (absolute) increase in efficiency from about 7 - 9%, much lower than state-of-the-art (Cu-poor) CuInSe₂, as also shown in Table 1.

To try to summarize my concerns again, the authors in my opinion decorate a very valuable, detailed and interesting study on the surface and defect chemistry of chalcopyrite thin films grown near stoichiometry with statements about boosting efficiency, technological impact that imply that the findings directly lead to an improvement of efficiency of CuInSe₂ solar cells, which I think is not warranted. The manuscript would be much more consistent and convincing if these statements were not made.

As a suggestion, the statements could be changed to

"Chemical instability of chalcogenide surfaces: impact on chalcopyrite devices well beyond the surface"
" Tailored chemical treatments are shown to mitigate anion vacancy defect formation."

"The device performance parameters of the various samples are summarized in Table 1. "

COMMENTS TO AUTHOR:

Reviewer #1:

The authors have responded to the raised concerns of the review and greatly improved the readability and consistency of the manuscript. I remain at odds with one point: the title claiming “technological impact” and the statement “Tailored chemical treatments mitigate anion vacancy formation and boost the efficiency of CuInSe₂ solar cells.” in the abstract, as well as the statement related to table 1 “the technological impact of the findings I summarised in Table 1, where all the device performance parameters of the various samples are reported”

To my understanding the authors explain in detail phase transformations and defect formation as well annihilation occurring at the surface of chalcopyrite thin films, for films with compositions close to the edge of the single phase region. With respect to devices or technology, they confirm (and explain) what has been done for Cu(In,Ga)Se₂ device manufacturing for many years, the utilization of Cu-poor compositions for high performance solar cells. To my knowledge they do not have to be KCN etched to yield high performance. The best CuInSe₂ solar cells to my knowledge are close to 14%, somewhat larger than the devices presented in table 1 (not discussed by the authors), and I do not see that the treatments proposed in this study would yield to a “boost in efficiency” of solar cells. In that sense “tailored” seems also not correct, since the best solar cells in table 1 are the PW and PB treatments, two different etchings yielding almost identical efficiencies.

Of course, if the “boost in efficiency” statement is related to the increase in efficiency of Cu-rich grown CuInSe₂ (as I think it is) then this becomes misleading to the reader, since the statement relates to a moderate (absolute) increase in efficiency from about 7 – 9%, much lower than state-of-the-art (Cu-poor) CuInSe₂, as also shown in Table 1.

To try to summarize my concerns again, the authors in my opinion decorate a very valuable, detailed and interesting study on the surface and defect chemistry of chalcopyrite thin films grown near stoichiometry with statements about boosting efficiency, technological impact that imply that the findings directly lead to an improvement of efficiency of CuInSe₂ solar cells, which I think is not warranted. The manuscript would be much more consistent and convincing if these statements were not made. As a suggestion, the statements could be changed to “Chemical instability of chalcogenide surfaces: impact on chalcopyrite devices well beyond the surface” “ Tailored chemical treatments are shown to mitigate anion vacancy defect formation.” “The device performance parameters of the various samples are summarized in Table 1. ”.

The referee understood very well the scope of our manuscript. As he rightly guesses, it is not our intention to mislead the reader by implying that the observed boost in efficiency is related to Cu-poor chalcopyrite. Indeed, we refer to Cu-rich chalcopyrite and throughout the manuscript we stress how the chalcopyrite Cu content impacts the inherent surface chemical properties of the semiconductor, which in turn have an impact on the performance of the photovoltaic devices based on chalcopyrite (hence more generally on the chalcopyrite PV technology). We also thank the referee for his/her concrete edit suggestions that help to fix any misunderstanding. We propose the following wordings, also taking into account formatting constraints and suggestions from the editorial office:

Title: Chemical instability at chalcogenide surfaces impacts chalcopyrite devices well beyond the surface

This wording is perfectly in line with the referee’s suggestion and avoids the use of punctuation.

Abstract: This study shows how selective defect annihilation is attained with tailored chemical treatments that mitigate anion vacancy formation and improve the performance of CuInSe₂ PV cells.

This wording acknowledges the referee’s concern of potentially misleading the readers that the study shows new world record efficient devices. At the same time, it stresses on the “how” the proposed tailored chemical treatments effectively improve the performance of chalcopyrite devices with respect to absorbers not subject to those treatments, which is the core of our study.

Text: The impact of the findings on solar cell performance is summarised in Table 2, where all the device parameters of the various samples are reported.

This wording is in line with the referee’s suggestion, as there is no mention of technological impact.